# Exploring and Exploiting Stability in Latent Flow Matching

**Rania Briq** [1 2]  **Michael Kamp** [2 3 4]  **Ohad Fried** [5]  **Sarel Cohen** [5]  **Stefan Kesselheim** [1 6 7]

## Abstract

In this work, we show that Latent Flow-Matching (LFM) models are robust to different types of perturbations, including data reduction and model capacity shrinkage. We characterize this stability by these models' tendency to generate similar outputs under identical noise seeds. We provide a perspective relating this phenomenon to flow matching theory, which indicates that this stability is inherent to the FM objective. We further exploit this stability to derive practical algorithms for more efficient training and inference. Concretely, first, we show that by training LFM models on significantly reduced datasets, performance is preserved, and in compute-constrained regimes, the model converges faster while maintaining quality. This yields multiple advantages, including savings in the training time due to faster convergence, and alleviating annotation effort when training conditional models. Second, LFM stability under architectural shrinkage gives rise to a two-model coarse-to-fine approach, one using a light-weight architecture for the first phase of the FM trajectory, and one with higher capacity for the second, thereby reducing the inference cost substantially. To determine which samples are informative, we introduce three sample-scoring criteria and evaluate them under standard metrics for generative models. Our results are thoroughly evaluated on multiple datasets, demonstrating the practical advantage of this stability, including data savings and a more than two-fold inference speedup while generating comparable outputs.

## 1. Introduction

Diffusion models (Song & Ermon, 2019; Song et al., 2020; Ho et al., 2020) have become a dominant paradigm for content generation across various modalities, such as images, videos, or medical imaging (Rombach et al., 2021; Popov et al., 2021; Blattmann et al., 2023; Webber & Reader, 2024). Motivated by the need for faster and more efficient sampling, recent work has explored Flow Matching (FM) (Lipman et al., 2022; Liu et al., 2022) as an alternative perspective for diffusion-based models. FM learns a time-dependent velocity field and offers a deterministic formulation for sample generation by solving an ordinary differential equation (ODE) rather than a reverse-time stochastic differential equation (SDE), often reducing the number of sampling steps and leading to faster generation.

Despite their astonishing success in generative modeling, training these models remains expensive: it requires large datasets, massive compute, long training times, and in conditional setups, extensive annotations. This naturally raises the question of whether dataset size, or even model capacity, can be reduced without sacrificing quality. In this work, as part of answering this question, we study latent flow-matching (LFM) models and provide substantial empirical evidence that their transport trajectories are strikingly stable under major perturbations, including dataset subsampling (pruning), architectural changes, and altered training configurations. One concrete instance that manifests this stability is when two models trained on disjoint subsets of the data frequently produce highly similar generations under identical noise seeds.

This invariance is not obvious in the context of a distribution learning problem. Related observations, however, have been reported in prior work: For score-based diffusion models, Kadkhodaie et al. (2024) observe consistent denoised trajectories across disjoint data splits and argue that models trained on different splits converge to similar harmonic bases in pixel space; a complementary perspective arises from connecting score-based diffusion models to entropic optimal transport, or Schrödinger bridges, which are known to be stable under perturbations of the marginals, such as those induced by data subsampling (Ghosal et al., 2022). However, prior work has primarily focused on score-based objectives applied to low-resolution images in pixel

---

[1] Forschungszentrum Jülich [2]Technical University Dortmund [3]Lamarr Institute [4]Institute for AI in Medicine, University Hospital Essen [5]Reichman University [6]Helmholtz AI [7]University of Cologne. Correspondence to: Rania Briq <r.briq@fz-juelich.de>.

*Proceedings of the 43rd International Conference on Machine Learning*, Seoul, South Korea. PMLR 306, 2026. Copyright 2026 by the author(s).

space, and has not demonstrated how such invariance can be exploited for principled methods to improve training and inference efficiency. We establish analogous stability in LFM across a broader range of perturbations, and translate it into practical algorithms for more efficient training and faster inference. A central question in our work is how to identify important samples and their number in a way that maintains performance. We find that thanks to LFM stability, even heavy dataset pruning does not cause performance degradation. For sample selection, we extend data pruning heuristics from discriminative models (Coleman et al., 2019; Paul et al., 2021; Mirzasoleiman et al., 2020; Abbas et al., 2024; He et al., 2024) to flow matching by reformulating their scoring functions along shared noise paths and timesteps in the FM objective. Specifically, we consider three criteria based on gradient signal, loss signal, and distribution coverage using clustering. We show that such pruning strategies can accelerate LFM training and improve standard generative metrics under limited compute budgets.

Our interpretation builds on the analyses of the closed-form solution to FM (Gao & Li, 2024; Bertrand et al., 2025). In particular, these works show that FM trajectories are largely shaped by individual training samples based on the softmax weights in this closed-form solution. This would indicate, as we empirically confirm, that pruning samples leaves most transport trajectories largely unchanged, provided the retained data remains sufficiently representative, and show how this improves training efficiency. We further leverage stability across model capacities for a two-stage coarse-to-fine generation strategy. In this approach, a lightweight model transports an initial noise sample during the first phase of the trajectory, while a higher-capacity one continues in a second phase where detail is enhanced, resulting in substantial inference speedups while maintaining generation quality. During training, we introduce a lightweight tailored fine-tuning procedure for improving the alignment at the intersection of both stages, since stability suggests that the learned velocity fields of both models are already closely aligned. We further combine this approach with a clustering-based pruning heuristic that balances the dataset distribution when training the coarse model of the first stage, and show that a balanced dataset is more critical for best performance than the sheer amount of data. See Fig. 1 for an overview.

We summarize our contributions as follows:

- We show that LFM exhibits striking stability: independently trained models converge to highly consistent transport trajectories under dataset subsampling, architectural changes, and different training configurations, including random seeds and FM sample-target coupling. We link this phenomenon to FM theory.
- We introduce three data pruning criteria suitable for FM

models, and study their influence across various metrics on different datasets. For example, on ImageNet, up to 75% of the data can be pruned without harming the performance. This reduction can accelerate LFM training and improve performance under a limited compute budget.
- Using trajectory stability, we propose a coarse-to-fine model, achieving $\approx 2.15\times$ faster inference while still maintaining quality. [1]

## 2. Methodology

### 2.1. Preliminaries

**Flow Matching.** FM as defined in Lipman et al. (2022); Liu et al. (2022) learns a time-dependent velocity field $v_\theta(x, t)$ parametrized by $\theta$, whose flow transports a simple source distribution $p_0$ (we choose $\mathcal{N}(0, I)$) towards a target distribution $p_1$. Let $x_0 \sim p_0$ and $x_1 \sim p_1$. Sampling integrates the ODE

$$\dot{x}_t = v_\theta(x_t, t), \quad x_0 \sim p_0, \quad t \in [0, 1],$$

to obtain $x_1 \sim p_1$. $v_\theta$ is trained using rectified flow by regressing the velocity on straight paths defined for a coupling between $p_0$ and $p_1$ defined by $x_t = (1 - t)x_0 + tx_1$ and $u(x_0, x_1, t) = \dot{x}_t = x_1 - x_0$ by minimizing the objective

$$\mathcal{L}(\theta) = \mathbb{E}_{\substack{x_0 \sim p_0, x_1 \sim p_1, \\ t \sim [0,1]}} \|v_\theta(x_t, t) - u(x_0, x_1, t)\|^2. \quad (1)$$

**Latent Flow Matching.** Similar to Latent Diffusion Models (Rombach et al., 2021), we perform FM in the latent space learned by a vector-quantized variational autoencoder (VQ-VAE) (Van Den Oord et al., 2017), which produces a lower-dimensional representation of the input data. In our setup, the images are encoded by the encoder, yielding a vector $x \in \mathbb{R}^{4 \times 32 \times 32}$ that represents samples from the target distribution $p_1$. At inference, we sample $\hat{x}_1$ by integrating the FM ODE and then decode the result to pixels using the decoder.

**FM Stability through the closed-form lens.** Rectified Flow (RF) admits a closed-form formula for the velocity field (Gao & Li, 2024; Bertrand et al., 2025) given by

$$\hat{u}^*(x, t) = \sum_{i=1}^{n} \lambda_i(x, t) \frac{x^i - x}{1 - t},$$

$$\lambda_i(x, t) = \text{softmax}_i\left(-\frac{\|x - tx^i\|^2}{2(1 - t)^2}\right), \quad (2)$$

---

[1]The preliminary idea of this work appeared in our workshop paper (Briq et al., 2026). This paper extends it substantially by providing a deeper theoretical analysis of the phenomenon, while demonstrating it on multiple datasets using more extensive pruning criteria and evaluation. It further proposes an inference acceleration approach that exploits stability.

(A) Transport stability under pruning

(B) Coarse-to-Fine training and inference

*Figure 1.* Overview of data pruning for efficiency and a *coarse-to-fine* model for inference speedup. **Top**: (Left) CelebA-HQ samples using the first two PCA components (blue), cluster centroids (orange) where the circle size is proportional to the cluster size. Pruning by balanced clustering ($\mathcal{C}_b$) equalizes the cluster sizes. (Middle) FM model transport for ten samples in PCA space. The full and pruned models lead to different trajectories only for the blue sample, whereas for all other source points, both models produce very similar trajectories. (Right) An FM model trained on data reduced by 50% using $\mathcal{C}_b$ (bottom row) generates strikingly similar images to a model trained on the full dataset. **Bottom**: Coarse-to-Fine approach. We use pruned subset $S'$ to train a light *Coarse* model, given a high-capacity pretrained *Fine* model. *Coarse* is trained to receive an intermediate latent $x_{t_0}$, obtained by integrating *Fine*'s velocity field $u_f(x, t)$ backward from $x_1$ to the $x_{t_0}$ (ODE inversion). At inference, *Coarse* forms the trajectory along $t \in [0, t_0)$ while *Fine* refines the details and texture along $t \in [t_0, 1]$. Stability indicates both models' trajectories can be smoothly stitched at the seam point $t_0$.

where $n$ denotes the dataset size. The weights form a softmax over the training samples indicating the contribution of each scaled training sample $tx_i$ to $x$. The objective in eq. 1 trains $v_\theta(x, t)$ to match the conditional mean over the target velocity whose optimum is the conditional mean $v^*(x, t) = \mathbb{E}[u(x_0, x_1, t) \mid x_t = x]$, which equals $\hat{u}^*(x, t)$ in eq. 2 for a finite dataset. Bertrand et al. (2025) have observed that the softmax in this formula becomes peaked in the early phase of the transport, indicating that one training sample $x^i$ dominates. This suggests stability in the optimal solution as long as these dominating samples are retained, motivating our experiments on probing and quantifying the sensitivity of the FM closed-form solution to sample removal in Sec. 2.2.

Notably, any model reproducing $\hat{u}^*(x, t)$ exactly would generate samples from the training set. Neural networks provide a smooth approximation of this function, hence can generate novel samples.

### 2.2. Flow Matching Stability under Dataset Pruning

**Closed form stability under pruning.** We examine the sensitivity of the closed-form transport for a finite training set to sample removal by removing fractions of the data randomly and comparing the transport induced by the full versus the pruned dataset. Our motivation is that a network optimized using the same objective can be expected to exhibit similar robustness behavior, allowing us to train LFM models on reduced datasets and split the transport trajectory across models. To quantify stability, we define an *assignment* met-

ric: for each trajectory starting from $x_0$, we compute its velocity field for $t \approx 1$ using eq. 2 (to avoid dividing by 0), and evaluate at which training sample the trajectory ends. We run the experiment on two datasets using 1000 noise samples $x_0$, namely CelebA-HQ, and a synthetic dataset (Synth), in order to verify that this robustness is not limited to a specific dataset. The synthetic data is generated by a Gaussian Mixture Model (GMM) with dimensionality $d = 4096$. For each pruning fraction, we calculate the percentage of samples whose assignment did not change given that the assigned sample based on the full dataset was retained, and report the result in Fig. 2. The assignment changes for only a small fraction of the samples (less than 3%) for any pruning fraction $pr$ up to 0.9.

This finding is consistent with the observation that $\hat{u}^*$ is often dominated by a small number of samples along the trajectory (Gao & Li, 2024; Bertrand et al., 2025). It highlights the fact that for a given dimensionality, despite perturbations that are most influential at the start of a given trajectory where samples contribute uniformly, the endpoint of the trajectory is not affected. We explain this by stability not being determined by pointwise equality in the velocity fields, but rather by the transport they induce along the whole trajectory. Removing a dominating sample can change an instant velocity, but if nearby samples from the same region remain, it can guide the trajectory toward a similar endpoint. Our path-deviation results (cf. Fig. 8 in the Appendix) support this interpretation: despite sample removal, the integrated trajectories remain close in high-dimensional latent space. In

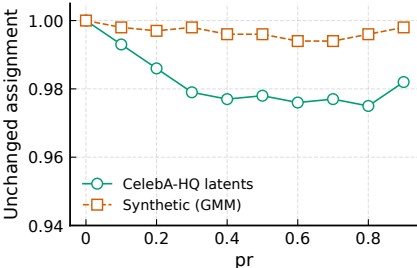

*Figure 2.* Fraction of samples transported with the closed form FM whose assignment of source points $x_0$ and training samples $x_1$ does not change, plotted as a function of the pruning fraction.

low-dimensional data, assignment changes are much more frequent, suggesting that intuitions from 2D examples do not necessarily transfer to high-dimensional latent spaces (cf. Fig. 9 in the Appendix).

## 2.3. Pruning methods

Given a training dataset $S$, we aim to find a subset $S' \subset S$ that optimizes the trade-off between computation and model performance. We consider three pruning criteria, adapting ideas from discriminative models to fit flow-matching dynamics (Paul et al., 2021; Abbas et al., 2024). We (i) rank based on a sample's loss computed along shared noise paths and timesteps; (ii) analogously rank based on a sample's *gradient norm* and (iii) cluster samples using their semantic features in a pretrained embedding space. For each method, we also apply the inverse criterion, i.e. we select samples with the lowest scores instead of the highest ones, and denote it by the superscript $-1$. For comparison, we also apply *random* sampling as a baseline.

*Gradient/Loss-based scoring ($\mathcal{G}/\mathcal{L}$).* To obtain the gradient- and loss-based pruning criteria denoted by $\mathcal{G}$ and $\mathcal{L}$, we train a small surrogate model on $\approx 7\%$ of the full training schedule. We use this model to estimate per-sample loss $\ell$, using $M = 2$ fixed random noisy samples and $T = 8$ timesteps, creating shared noise paths for all the samples and decreasing the variance stemming from randomness. The values are then averaged over $M$ and $T$ using exponential moving average (EMA) estimate per $t_k$, which we maintain during computation and update at each iteration, to reduce variance.

$$s_i^{\mathcal{G}} = \frac{1}{T} \sum_{k=1}^{T} \frac{1}{M} \sum_{m=1}^{M} \frac{\left\| \nabla_\theta \ell(x_i; t_k, x_0^{(m)}) \right\|_2^2}{\mu_g(t_k)},$$

where $x_i \in S$, $x_0^{(m)} \sim p_0$ and $\mu_g(t_k)$ is the per-$t$ mean of the gradient norm $\|\nabla_\theta \ell\|_2^2$. $s_i^{\mathcal{L}}$ is defined similarly, replacing $\|\nabla_\theta \ell\|_2^2$ by $\ell$ and $\mu_g$ by $\mu_\ell$. Since the gradient ($\mathcal{G}$) computation is costly, we only use it to study the impact of high-gradient samples in FM training.

*Clustering-based scoring ($\mathcal{C}$).* To obtain the clusters using this criterion, we apply k-means (Lloyd, 1982) in CLIP (Radford et al., 2021) image embedding space using cosine distance.

There are two criteria to consider here: (i) how many samples to select from a cluster, and (ii) which samples. For (i), we implement a *proportional* and *balanced* approach, selecting either a number proportional to the cluster size or an equal number from each cluster. The first inherits the underlying distribution imbalance, while the latter balances skewed datasets. For (ii), we rank a cluster's population based on either (a) each sample's distance from the cluster center, and select either those located nearest or furthest from its center, or (b) kernel selection that greedily chooses samples such that the mean of their Gaussian kernel in the latent space matches that of the whole cluster, or (c) coreset selection, in which the sample furthest from the current selected set is repeatedly added. Further details in Appendix A.1. The nearest samples form a subset that retains the core characteristics of the distribution, while the furthest samples cover more scarce samples. In contrast, kernel-based selection preserves the overall cluster representation while coreset greedily optimizes over cluster coverage by spreading samples apart. We refer to these variants as $\mathcal{C}_{p/b}^{1/-1/\kappa/\text{cs}}$, with $p/b$ indicating *proportional/balanced*, and $1/-1/\kappa/\text{cs}$ indicating nearest/furthest/kernel-based/coreset selection. A global variant of kernel-based and coreset selection is also implemented, where the optimization is done across the entire dataset rather than per cluster.

**Preprocessing cost.** Clustering requires a single forward pass over the dataset for feature extraction, followed by k-means in CLIP embedding space, and is therefore incurred only once before training. In contrast, training requires repeated forward and backward passes over many iterations. Thus, when training on a reduced dataset reaches a target quality with fewer iterations, this one-time cost can be offset by the training-time savings. Furthermore, preprocessing does not require annotations, thereby reducing the annotation effort when training on reduced datasets. A reduced dataset also lowers the data-loading overhead during training.

## 2.4. Coarse-to-Fine model (C2F)

The stability phenomenon that FM models exhibit has inspired our *coarse-to-fine* model approach, which aims to reduce inference cost. In this design, we train two models, where the first is light-weight and evolves the first part of the trajectory, while the second is high-capacity and evolves the remaining part. In our setup, *Coarse* is trained on a subset of the data pruned based on the proposed balanced clustering method $\mathcal{C}_b$, using the DiT-S/2 architecture (33 M

parameters), and covering the first interval $t \in [0, t_0)$ of the trajectory, while *Fine* is pretrained on the full dataset, using DiT-XL/2 architecture (675M parameters) and covering $t \in [t_0, 1)$. To make a smooth transition at the seam between the two models, we finetune *Coarse* for a few epochs. In addition to the FM loss along $t \sim \mathcal{U}([0, t_0))$, we add a continuity loss term at $t_0$ for $\hat{x}_{t_0}$. To encourage Coarse's predictions to be close to Fine's, we compute $\hat{x}_{t_0}$ by applying ODE inversion using the fine model: starting at the training point latent $x_1$, we integrate the ODE induced by *Fine* backward in time from $t = 1$ to $t_0$ using the Euler method:

$$x_{k+1} = x_k + h\, v_F(x_k, t_k), \quad t_{k+1} = t_k + h, \quad h < 0, \quad (3)$$

where $v_F(x, t)$ is the fine model's estimated velocity field and $h$ is a tiny step. This produces $x_{t_0}$ that lies on *Fine*'s trajectory, at which point *Coarse* continues. The continuity term matches both models' outputs at the seam, denoted by $v_m(x_{t_0}, t_0)$ for $m \in \{(F)ine, (C)oarse\}$.

$$\mathcal{L}_{\text{coarse}} = \mathbb{E}\, \mathcal{L}_{\text{FM}}^{t \in [0, t_0)} + \lambda_v\, \mathcal{L}_{\text{seam}}^v,$$
$$\mathcal{L}_{\text{seam}}^v = \|v_F(x_{t_0}, t_0) - v_C(x_{t_0}, t_0)\|^2. \quad (4)$$

The stability property enables a nearly smooth transition even without fine-tuning, since both models' trajectories at the intersection point are close, as indicated by the closed-form analysis. In fine-tuning, we only need to stitch the models together by a seam loss to encourage continuity. At inference, the coarse model performs most of the denoising through $t_0$ using a light architecture, and the fine model takes over for $t > t_0$ using the high-capacity model only for a small portion of the trajectory.

# 3. Experiments

**Datasets.** We conduct our experiments on CelebA-HQ (28k train / 2k val) (Karras et al., 2017), FFHQ (63k train / 7k val) (Karras et al., 2019), and ImageNet-1K (1.2M train / 50k val, 1000 classes) (Russakovsky et al., 2015).

**Experimental Setup.** We use the transformer-based architecture DiT (Peebles & Xie, 2023), and replace diffusion with flow-matching transport (Esser et al., 2024). Following DiT, we use the EMA model for inference. We also train a vector-quantized variational autoencoder (VQ-VAE) (Van Den Oord et al., 2017), which encodes the images into $4 \times 32^2$ latents. To prevent leakage from other datasets, we train the VAE using the same target dataset.

On CelebA-HQ and FFHQ, we trained an unconditional model based on DiT-S/2 and DiT-B/2 architectures respectively, unless stated otherwise. On ImageNet, we train a label-conditioned model based on the DiT-XL/2 architecture. For the architectural change experiment, we additionally train a U-Net backbone (Ronneberger et al., 2015), which is

a CNN-based architecture. Further details are provided in Appendix C.[2]

**Evaluation Metrics.** We report FID (Heusel et al., 2017), and provide $\text{FD}_{\text{CLIP}}$ and Precision/Recall (fidelity/coverage) (Kynkäänniemi et al., 2019) in the appendix. For quantifying FM stability on CelebA-HQ and FFHQ, we measure ArcFace cosine similarity for faces (Deng et al., 2019), a standard embedding model for face identification. The similarity is computed in a pairwise manner between images starting from the same $x_0$. We denote the mean cosine similarity by the symbol $s$. For reference, the average similarity of random pairs of images on CelebA-HQ is $s = 0.37$. On ImageNet, we quantify the stability using cosine similarity on DINO features (Oquab et al., 2023). $N$ denotes the number of generated samples, and in *CelebA-HQ* and *FFHQ* experiments, $N = 4k$, and in *ImageNet*, $N = 10k$, unless stated otherwise. The pruning fraction ($pr$) denotes the fraction of pruned samples, i.e. the retained fraction is $1 - pr$. We refer to a model trained on a pruned dataset/full dataset as *pruned/unpruned* for brevity, but it does not mean we prune the model.

## 3.1. Results on CelebA-HQ

**Generative evaluation.** We apply the proposed pruning methods and their inverse for pruning fraction $pr = 0.5$, producing disjoint subsets for each method-inverse pair, and train an LFM model for each case. Fig. 3 includes qualitative samples, the corresponding FID values at $pr = 0.5$ and the FID curve across pruning fractions, all trained using the same budget. In the visual results, the generated samples are grouped by the initial noise $x_0$. Despite the models being independently trained on different or even disjoint training subsets, images starting from the same $x_0$ converge to visually similar images. Even $\mathcal{G}^{-1}$ and $\mathcal{L}$, for which FID is considerably worse and visual artifacts are apparent, perceptually the images still appear similar to those of the inverse method.

The FID table in Fig. 3 for $pr = 0.5$ shows that $\mathcal{C}_b$, $\mathcal{C}_b^\kappa$ and $\mathcal{L}^{-1}$ slightly improve the unpruned model, with $\mathcal{C}_b^\kappa$ performing best. We additionally evaluate two coreset baselines at $pr = 0.5$: a global coreset yields FID 26.25, and the balanced clustering variant $\mathcal{C}_b^{cs}$ yields FID 26.94, both performing worse than $\mathcal{C}_b$. This is consistent with the behavior of $\mathcal{C}_b^{-1}$, which also prioritizes distant samples and shows degraded performance.

The FID-vs-pr plot in Fig. 3 shows that $\mathcal{C}_{b/p}$, $\mathcal{G}$ and $\mathcal{L}^{-1}$ preserve or slightly improve FID up to $pr = 0.5$. At higher pruning fractions, FIDs deteriorate for all methods. Selecting the lowest-grad or highest-loss samples ($\mathcal{G}^{-1}$ or $\mathcal{L}$)

---

[2]Code and implementation details are available at: https://github.com/briqr/explo-r-it-ing_lfm_stability.

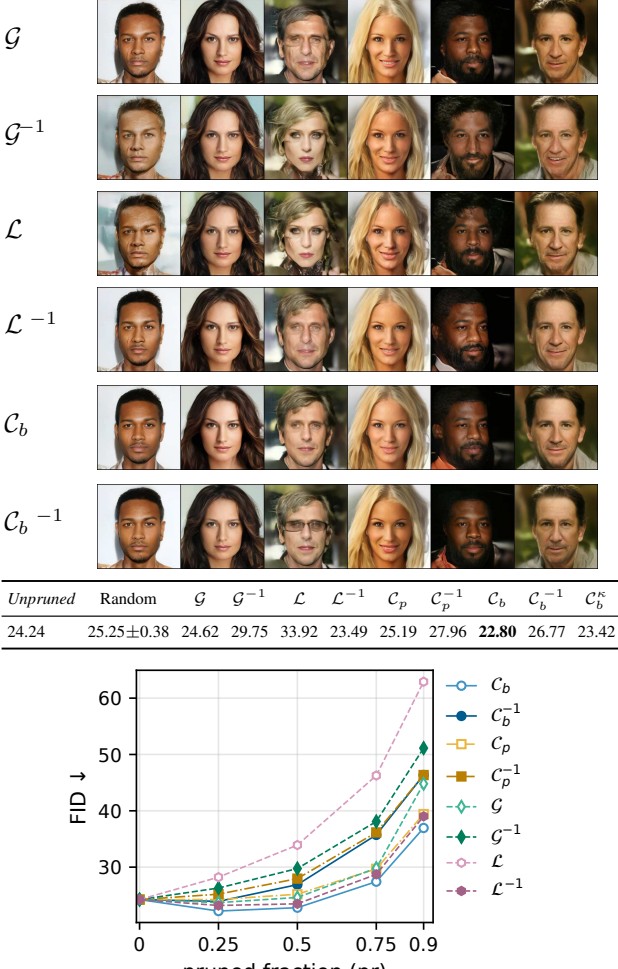

| Unpruned | Random | $\mathcal{G}$ | $\mathcal{G}^{-1}$ | $\mathcal{L}$ | $\mathcal{L}^{-1}$ | $\mathcal{C}_p$ | $\mathcal{C}_p^{-1}$ | $\mathcal{C}_b$ | $\mathcal{C}_b^{-1}$ | $\mathcal{C}_b^{\kappa}$ |
|---|---|---|---|---|---|---|---|---|---|---|
| 24.24 | 25.25±0.38 | 24.62 | 29.75 | 33.92 | 23.49 | 25.19 | 27.96 | **22.80** | 26.77 | 23.42 |

*Figure 3.* Stability and FID under different pruning criteria at $pr = 0.5$. **Top:** Images generated by independent models trained on different subsets of the data. The samples in each column are generated starting from the same $x_0$. **Center:** FID for each method. Random is averaged over three seeds. **Bottom:** FID on CelebA-HQ vs $pr$ for all pruning criteria.

substantially worsens FID. For highest-loss samples, its performance contrasts that of discriminative models (Paul et al., 2021), suggesting that in FM, loss-based scores should be interpreted differently. Specifically, high-loss samples may reflect erroneous velocity fields rather than useful hard examples, which is supported by the better performance of $\mathcal{L}^{-1}$.

**Stability quantification.** The ArcFace similarity $s$ for images of each pruning method with images by *unpruned* ranges between 0.79 and 0.83, while the similarity of each method with its inverse ranges between 0.72 and 0.74. For examining stability across random seeds (indicating different initialization and training stochasticity), we train two models using two different random seeds and obtain $s = 0.80$. (For a detailed quantification, see Appendix A.3).

## Stability tests.

Our findings regarding model stability motivated numerous experiments stress-testing this observed behavior, with visual results shown in Fig. 4. The first test is related to architectural modifications, where in one variant we increase the model capacity from DiT/S-2 (33M, 12 layers), which is our reference model, to DiT/XL-2 (675M parameters, 24 layers). In another, we switched to a U-Net architecture. We report qualitative results in Fig. 4a. The substantial increase in transformer capacity ($20\times$ more weights) only leads to minor changes ($s = 0.81$), and a slight increase in quality (FID=24.24→22.87). For a U-Net architecture, the drop in similarity is clearly more visible with $s = 0.55$, however, the coarse attributes are preserved. This supports our conjecture that independent of the specific model architecture, the flow fields that are learned are highly similar and capture a similar global structure.

Fig. 4b shows a stronger dataset perturbation. Row 1-2 correspond to a mode removal test: We trained the model on images classified as female (row 1) or male (row 2) by PaliGemma (Beyer et al., 2024) in a zero-shot classification setting[3]. As expected, this disturbs stability: Samples $x_0$ originally corresponding to the removed cluster are transported to different latents $x_1$, while images corresponding to the retained cluster preserve similarity. The average similarity obtained is $s = 0.76$ and $s = 0.58$ respectively, suggesting changes in the flow field happen locally, while regions corresponding to the retained cluster are largely unaffected. The higher similarity for perceived female (F) compared to perceived male (M) is due to the gender imbalance in CelebA-HQ (67% F vs 33% M). In Fig. 4c, we use the VAE trained on *CelebA-HQ* and train only the FM model on *FFHQ* (FM$_{\text{FFHQ}}$), a different but same-domain dataset. Qualitatively, we observe that despite some clear divergences in appearance, image pairs matched by $x_0$ remain visually related ($s = 0.58$). This result indicates that when the latent space is fixed, a different but same-domain dataset still lies approximately on the same manifold, leading to learning similar trajectories.

In contrast, LFM is sensitive to changes in the latent space parametrization as seen in Fig. 4d. For example, modifying the latent space by changing the VAE, such as changing its training seed (VAE-swap), translates to a change in the images (row 1, $s = 0.32$). In row 2-3, we conduct a controlled change in the latent space by applying a simple invertible transform $A$ on the latent space during training, such that $x_1' = Ax_1$. The flow path then becomes $x_t = (1 - t)x_0 + tAx_1$. At inference, we decode the transform-inverted latent $\hat{x}_1$, i.e. $A^{-1}\hat{x}_1$ to undo the transform. In row 2, only $x_1[0]$, i.e. the first feature of

---

[3]Gender is modeled as binary for this study only as a technical simplification and is classified as perceived by the classifier.

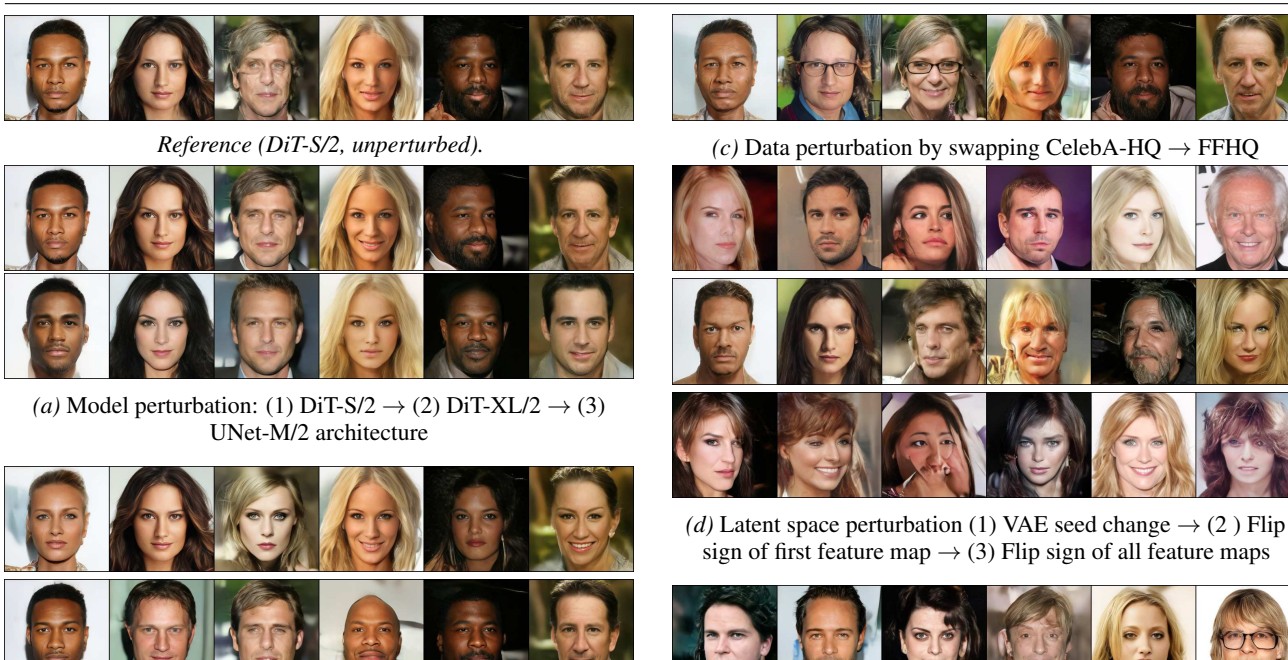

*Reference (DiT-S/2, unperturbed).*

*(c)* Data perturbation by swapping CelebA-HQ → FFHQ

*(a)* Model perturbation: (1) DiT-S/2 → (2) DiT-XL/2 → (3) UNet-M/2 architecture

*(b)* Data perturbation: (1) removal of perceived males, (2) females

*(d)* Latent space perturbation (1) VAE seed change → (2) Flip sign of first feature map → (3) Flip sign of all feature maps

*(e)* Algorithm perturbation: FM→ score-based diffusion

*Figure 4.* Stability under various perturbations. *(a)* (1) FM + VAE baseline; (2-3) Swapping DiT-S/2 with DiT-XL/2 and UNet-M/2 architecture. *(b)* (1-2) Removing a perceived-gender mode breaks stability for the removed cluster. *(c)* Swapping CelebA-HQ → FFHQ while using CelebA-HQ VAE preserves stability. *(d)* (1) Changing only the VAE training seed changes the output (VAE-swap); (2-3) Applying invertible transforms to the latent space (first/all features) changes the outputs to various degrees. *(e)* Using score-based diffusion instead of FM completely alters the outputs.

the latent (Latent-0) was sign-flipped, and similarities are perceptually retained, while in row 3, all feature maps were sign-flipped (Latent-all). These transforms clearly break stability ($s = 0.48$ and $s = 0.32$ respectively). This sensitivity to a change in the latent coordinate system suggests that LFM does not learn a coordinate-independent canonical mapping from a source noise to a latent data sample.

In Fig. 4e, we train a score-based diffusion instead of FM using the same architecture. This change of objective breaks stability; the model might have learned the same distribution as FM, but its mapping from noise to the latent space is distinct. This result indicates that trajectories are unique to the underlying objective, and that they are not inherent to the architecture's inductive bias alone, but rather of an interplay of objective and latent space, as a modification in either of these components resulted in completely changed trajectories too. Architecture plays a role too, but to a lesser degree as observed by the somewhat retained similarity when switching to a U-Net architecture.

These results indicate that the transport learned by LFM is not absolutely invariant to all perturbations, and can break when the latent coordinate parametrization is changed or when removing entire modes of the data distribution.

Although our experiments employ VQ-VAE, our results overall indicate that a discrete VQ-VAE latent space is not a prerequisite for stability. In Appendix A.4, we verify this by an additional perturbation experiment using a fixed continuous KL-VAE latent space and observe that stability holds under $\mathcal{C}_b$ pruning.

**Coarse-to-Fine (C2F)**. Fig. 5 shows that our proposed *C2F* approach reduces inference cost substantially while improving quality, yielding $\approx 2.15\times$ faster transport for $t_0 = 0.7$ (43.53 vs 93.95 ms/img) on an NVIDIA H100 GPU (batch=128, resolution $256^2$) compared to *Fine* alone. We vary the seam point $t_0$ and find $t_0 = 0.7$ strikes a good balance between FID and runtime. For *unpruned* (full dataset), degradation does not occur up to $t_0 = 0.4$, demonstrating that the early trajectory does not need a heavy model. For the pruned model, the improvement comes from the pruning itself despite $t_0$ increasing and Coarse performing more of the trajectory. As $t_0$ increases, indicating *Coarse* evolves a larger part of the trajectory, the inference cost decreases. To show that stability is crucial for this approach, we perform the C2F$_{male}$ experiment, in which we selected *Coarse* to be one of the variants that violated stability (cluster removal) (Fig. 4), specifically, *Coarse* is trained and finetuned only on a single perceived gender. In this case, the

performance degrades significantly (FID = 44.92). This shows that if stability does not hold across various architectures, fine-tuning with a seam loss does not suffice to align two divergent trajectories (cf. Fig. 18 for visual results.)

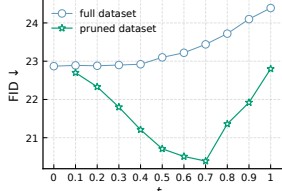 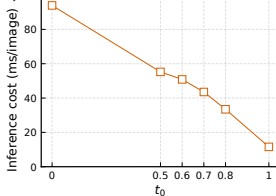

FID on CelebA-HQ vs $t_0$.     Inference cost (ms/image) vs $t_0$.

*Figure 5.* Performance of the *coarse-to-fine* approach on CelebA-HQ. For $t_0 = 0$, only *Fine* operates, while for $t_0 = 1$, only *Coarse* operates. (Left) In *C2F*, *Coarse* trained on a pruned dataset with $t_0 = 0.7$ yields the best performance in FID. For the full dataset, FID does not worsen until $t_0 = 0.5$. (Right) Inference cost vs. $t_0$.

**Balanced generation.** Since we have shown LFM exhibits stability, we exploit this property to mitigate biases in data distributions through balanced dataset pruning. In Table 1, we evaluate the effect of pruning on fairness (Zhang et al., 2023; Friedrich et al., 2023). Specifically, we report the KL discrepancies $D_{kl}$ between the generated images $\tilde{p}_1$ and a uniform distribution $q$ across multiple attributes given by $D_{KL}(\tilde{p}_1, q) = \sum_v p(v) \log \frac{p(v)}{q(v)}$. Here, $v$ denotes the attribute value, as classified by PaliGemma (Beyer et al., 2024)[4]. A lower distance indicates a more balanced generated distribution $\tilde{p}_1$. We observe that balanced clustering ($\mathcal{C}_b$), despite being attribute-agnostic, yields the lowest discrepancy across all attributes, also compared to *unpruned*, indicating that $\mathcal{C}_b$ mitigates bias in skewed datasets.

The label-aware clustering $(\mathcal{C}_b)_{\text{gender}}$ selects an equal number of samples from both genders' clusters, and the resulting model generates both genders nearly equally. In addition, we observe that this targeted pruning preserves FID (24.3 vs 24.24 for *unpruned*). This indicates that explicit label-aware balancing of the dataset provides best control over the distribution of generated samples with respect to a given attribute. The drawback, however, is that it requires an explicit choice of attributes to balance, and its computation becomes exponential in their number. In contrast, attribute-agnostic clustering $\mathcal{C}_b$ does not require this explicit design choice and can still mitigate biases across multiple attributes simultaneously, but without full control over individual attributes.

**ImageNet.** In addition to CelebA-HQ, we perform similar experiments by training conditional LFM models on ImageNet and its subsets. As in our previous results, we focus on $\mathcal{C}_b$, but for comparison, we also evaluate $\mathcal{C}_p$, $\mathcal{C}_b^{-1}$,

---

[4]We emphasize that these attributes are classified as perceived by the classifier. Our objective is to study and improve the representation across groups.

|  | Gender | Age | Skin-tone | Hair color |
|---|---|---|---|---|
| *Unpruned* | 0.044 | 0.608 | 0.123 | 0.844 |
| $\mathcal{G}$ | 0.043 | 0.690 | 0.104 | 0.779 |
| $\mathcal{L}^{-1}$ | 0.040 | **0.104** | 0.466 | 0.858 |
| $\mathcal{C}_p$ | 0.043 | 0.664 | 0.127 | 0.619 |
| $\mathcal{C}_b$ | **0.016** | 0.575 | **0.098** | **0.571** |
| $(\mathcal{C}_b)_{\text{gender}}$ | **0.005** | – | – | – |

*Table 1.* $D_{KL} \downarrow$ to a uniform distribution on CelebA-HQ for $\text{pr} = 0.5$. Lower indicates a more uniform representation across attribute values. $(\mathcal{C}_b)_{\text{gender}}$ prunes such that both genders' fractions are equal. $N = 4k$.

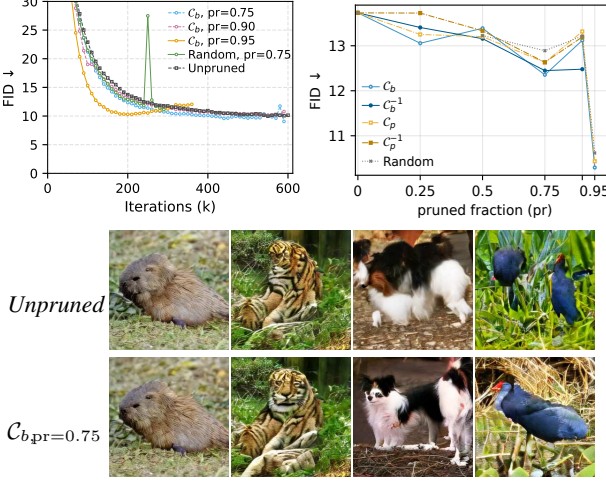

*Figure 6.* **Top left:** FID across training iterations. Pruning improves faster, extreme pruning (pr=0.95) at first performs best but degrades after $170k$, and pr=0.9 degrades after $590k$. *Unpruned* and $\mathcal{C}_{b,\text{pr}=0.75}$ both plateau at $\approx 600k$ as they reach the best FID. **Top right:** FID on ImageNet vs. pr at a fixed training budget of $200k$ iterations. **Bottom:** Qualitative ImageNet samples at the $200k$ iteration checkpoint.

$\mathcal{C}_p^{-1}$ and random sampling. We note that these experiments, being run on the DiT/XL-2 architecture, require significant computational resources; we therefore performed the longest runs only on $\mathcal{C}_b$. In Fig. 6 (top), we plot the FID for *unpruned* and $\mathcal{C}_b$ with $\text{pr} \in \{0.75, 0.9, 0.95\}$ as a function of the training iterations. $\mathcal{C}_b, \text{pr} = 0.95$ converges fastest at first, but the FID degrades after about $200k$ iterations. For $\text{pr} = 0.75$, a slight performance gain over *unpruned* is maintained throughout the training run up to $600k$, where the training progress slows down and the gap diminishes. $(\text{pr} \geq 0.9)$ demonstrates intermediate behavior, where a slightly faster early convergence levels off later during the training. At iteration $200k$, we observe the largest discrepancy between the models. Fig. 6 (middle) shows the FID at this point ($200k$) for various pr values. We observe that every method, including random sampling, improves over *unpruned* for all considered pr values. This quantitative finding is supported by the visual image quality. The lower panel of Fig. 6 displays four representative example images generated by models trained on the full dataset and the $\mathcal{C}_b$-pruned dataset at $\text{pr} = 0.75$. Similar to our stability findings

on CelebA-HQ, we confirm the qualitative and quantitative similarity of the images that start from the same $x_0$. The images generated by the model trained on the $\mathcal{C}_b$-pruned dataset appear sharper and exhibit clearly weaker artifacts than the *unpruned* model.

These results and observed behaviors on ImageNet differ from those obtained on CelebA-HQ. Specifically, on CelebA-HQ, we did not observe a considerable difference in convergence during training, rather, the training curves leveled off at a similar pace. As the size and diversity of ImageNet are significantly larger than those of CelebA-HQ, it is plausible that differences in convergence speed appear more pronounced in the generated samples.

The stability metric under dataset pruning remains high under DINO cosine similarity, ranging from 0.71–0.74 (std. $\approx 0.13$–$0.14$) over pairs across the different variants relative to *unpruned*. **FFHQ.** We have also conducted similar experiments and evaluations on the *FFHQ* dataset. The results are mainly consistent with CelebA-HQ in terms of stability and generative metrics, we therefore defer the results to Appendix A.6.

## 4. Discussion and Conclusion

In this work, we have shown that LFM models remain stable despite substantial perturbations to training. We characterized this stability through trajectory similarity: different models' trajectories initialized from the same source point $x_0$ follow paths that end in similar latent vectors $x_1$. We demonstrated that this similarity across models is preserved under (i) significant pruning of the training data, even when models trained on disjoint subsets, and (ii) changes in model size or architecture. Furthermore, we provide evidence that this stability is a characteristic of the rectified FM paradigm under fixed latent representations, rather than being tied to a specific training configuration. Systematically removing an entire mode of the distribution, as studied through pruning based on perceived gender, images corresponding to the remaining mode remain qualitatively unchanged. Only perturbations of the latent space structure or a complete change of the generative objective consistently broke this behavior. Furthermore, when probing the closed-form solution of FM, which guarantees exact recovery of the training data, we observe analogous behavior. In particular, identical source points $x_0$ are nearly always transported to the same training samples $x_i$, provided that the corresponding samples have not been removed.

To translate these stability properties into practically useful insights, we devised several pruning criteria that address the structure of the data in terms of clusters, as well as sample influence on training, measured by the magnitude of the corresponding loss and gradient. Among these prun-

ing methods, the strategy aimed at mitigating imbalances between clusters ($\mathcal{C}_b$) has proven to be the most effective. Our findings indicate that pruning with this approach can even improve the quality of generated images. For the comparatively small CelebA-HQ dataset, we observe a slight improvement in FID when pruning approximately 50% of the data. For the much larger and more diverse ImageNet dataset, we observe faster convergence during training. We conjecture that this behavior arises from the interaction of three effects. First, under a fixed compute budget, reducing the dataset size increases the effective number of training epochs, resulting in higher repeated exposure to each retained sample. Second, due to the inherent stability of LFM, this additional exposure does not lead to overfitting or trajectory collapse, but rather reinforces consistent transport behavior. Third, balancing the dataset across semantic clusters reduces redundancy and prevents overrepresented modes from dominating the learning dynamics. Taken together, these effects allow the model to allocate its capacity more uniformly across the data manifold, leading to improved image quality or faster trajectory stabilization. It is noteworthy that applying the pruning criteria incurs additional costs, but since they are invoked only once before training, their benefit in faster convergence offsets the overhead.

Our observation that pruning can accelerate training has practical implications. While the ImageNet dataset evaluated in this work is relatively homogeneous by construction, consisting of one million images distributed fairly evenly across 1000 classes, many practically relevant datasets, such as those obtained via web crawling, are likely to be highly skewed. We have also shown that pruning strategies aimed at balancing a dataset provide a mechanism to control and mitigate distributional imbalances in generated data, which is critical for fairness-sensitive applications.

Finally, we exploited trajectory stability to accelerate inference. Specifically, we split the trajectory into two phases, in which the first phase is realized using a light-weight model, while only a relatively small part of the trajectory is evolved using a larger model in the second phase. This was achieved by our stitching strategy based on backward integration of the ODE and the application of a seam loss. Limitations pertaining to this approach include the lack of automatic adaptation for the split point $t_0$, which we treated as a hyperparameter. Empirically, our results suggest that performance is not highly sensitive to this choice up to $t = 0.5$. In practice, finding $t = 0.5$ is inexpensive, since it can be tuned on a small validation set. We leave the finding of an adaptive criterion as a direction for future work. We also note that two models must be invoked at inference, but the additional memory is limited because the coarse model is substantially smaller than the fine model.

## Impact Statement

The robustness of latent flow-matching models to perturbations can be leveraged to reduce training and inference cost. Our analysis using perceived attributes such as gender and skin-tone demonstrated that balancing the training data encourages the generated distribution to be closer to a uniform one, while maintaining performance. However, perceived attributes depend on the classifier and its internal representation rather than ground-truth attributes. Furthermore, the ability to control the generation distribution through clustering can be used to steer the distribution unfairly either intentionally or unintentionally. We therefore encourage appropriate data and attribute scrutiny through a collaboration between technical and ethics experts when curating data for an intended downstream use-case.

## Acknowledgments

This work is funded by the German Federal Ministry for Economic Affairs and Energy within the project "nxtAIM". Additionally, the authors gratefully acknowledge the Gauss Centre for Supercomputing e.V. (www.gauss-centre. eu) for funding this project by providing computing time on the GCS Supercomputer JUWELS Booster at Jülich Supercomputing Centre.

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

# A. Further experiments and results

## A.1. Clustering

**Balanced clustering ($\mathcal{C}_b$).** In this method, when pruning based on a given pruning fraction $pr$, we divide the number of remaining samples equally by the number of clusters $k$: $\frac{(1-pr)\cdot|S|}{k}$ ($S$ denotes the dataset). If some clusters are too small to supply this number, we distribute the missing samples across larger clusters. By default, $\mathcal{C}_b$ selects samples located nearest to the center.

**Balanced clustering using kernel-based sampling ($\mathcal{C}_b^\kappa$).** Given the semantic clusters, we want to select samples equally from each cluster. To determine which samples to select, the kernel method selects samples greedily of those whose mean is closest to the full cluster's mean in the latent space (Chen et al., 2012). Specifically, for a cluster $C_l$ with samples $\{x_i\}_{i=1}^{|C_l|}$, we define its mean as

$$\mu_{C_l} = \frac{1}{|C_l|} \sum_{i=1}^{|C_l|} k(x_i, \cdot),$$

where $k(\cdot, \cdot)$ denotes a Gaussian (RBF) kernel computed on the latent representations. We then select a subset of samples whose empirical kernel mean best approximates $\mu_{C_l}$. The Gaussian kernel computation is approximated using Random Fourier Features (Rahimi & Recht, 2007) to make the computation feasible. As reported earlier, the $\mathcal{C}_b$ criterion yielded the best FID on CelebA-HQ for $pr = 0.5$ (FID = 22.8), followed closely by this kernel-based criterion (FID=23.42). If we make the kernel-based selection global over the whole dataset rather than constraining it to clusters, the FID decreases slightly to 24.82, highlighting the importance of semantic-based selection.

**Balanced clustering using coreset-based sampling ($\mathcal{C}_b^{cs}$).** This is a coreset-style criterion inspired by Gonzalez (1985); Sener & Savarese (2017), which applies farthest-point traversal inside each cluster: starting from an initial sample, we repeatedly add the sample whose distance to the current selected set is maximal, where distance to a set is measured by the distance to the nearest sample. The degraded FID results obtained under this strategy (FID = 26.94) suggest that for LFM training, selecting samples closer to the center is more effective than maximizing geometric diversity alone.

**Ablative experiment on the number of clusters.** In our experiments, we determined the number of clusters $k = 24$ based on the elbow method. In Fig. 7, we report $\text{FD}_{\text{CLIP}}$ as a function of the number of clusters $k$ for $pr = 0.70$ using $\mathcal{C}_b$, i.e. we train on 30% of CelebA-HQ. While $k = 2$ achieves the best FD, we found that the split concentrated on perceived gender despite it being label-agnostic. However, with $k = 2$, the diversity would be limited to one attribute only, whereas a higher number of clusters encourages more diverse attribute selection.

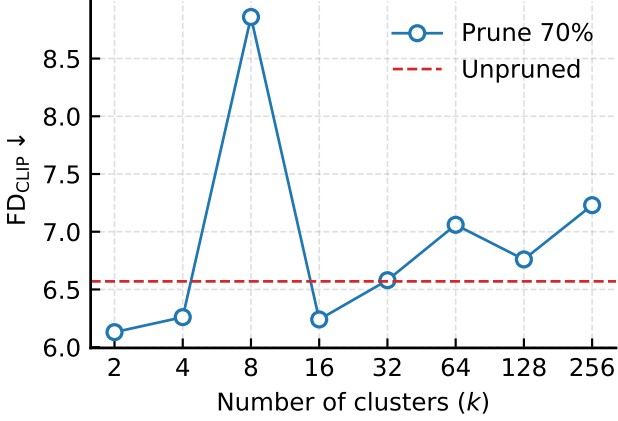

*Figure 7.* $\text{FD}_{\text{CLIP}}$ vs. the number of clusters $k$

**Ablative experiment on gender.** We further probe the gender balancing experiment by devising multiple experiments that generate an equal number of samples for each gender. We can (i) train two separate models (female-only and male-only) and generate both genders equally using the respective model, (ii) train a single model on a balanced dataset of both genders (the

| Model variant | generated perceived Female/Male fraction | $FD_{CLIP}$ ↓ | Training iterations [k] ↓ |
|---|---|---|---|
| Male-only + female-only (two models, mix samples) | 50/50 | 6.25 | 50 each |
| Male–female balanced (single model) | 55/45 | 5.62 | 80 |
| Oversampling | 55/45 | 5.40 | 120 |
| Whole dataset (original) | 67/33 | 5.29 | 140 |

*Table 2.* CelebA-HQ ablation on controlling gender generation composition. We report $FD_{CLIP}$, which is also consistent with FID.

earlier experiment that we referred to as $(\mathcal{C}_b)_{gender}$, and (iii) oversample the male images during training so both genders' clusters are equal. For comparison, we also train on the original imbalanced dataset (male 33%, female 67%).

For this experiment, we report $FD_{CLIP}$ in Table 2. We obtain $FD_{CLIP}$ that is comparable across all of them, with the distinction in the training runtime, where $(\mathcal{C}_b)_{gender}$ converged fastest at $80k$ iterations. While the separate female and male models converged earlier at $50k$, their cumulative training iterations are $100k$.

**Fairness experiment**   In the fairness results reported in Table 1, to obtain these attribute values, we feed the images generated by our various models into PaliGemma VLM (Beyer et al., 2024) and prompt it to assign a label based on a given list of labels for each attribute. For example, for the hair-color attribute, we use the prompt:

```
"Hair color":  "What is the hair color of the person in the image?  (e.g., black,
brown, blonde, red, gray, white, or a mix)"
```

We note that the fairness results that we obtained via $\mathcal{C}_b$, which is label-agnostic, are weaker than label-aware clustering in terms of fairness (such as $(\mathcal{C}_b)_{gender}$). However, complete control over the distribution of generated labels requires an exponential number of clusters. For example, if we want to equalize across *gender*, *skin-tone* and *age* simultaneously, we first need to create $|\mathcal{V}_{gender}| \times |\mathcal{V}_{skin-tone}| \times |\mathcal{V}_{age}|$ equalized clusters, where $|\mathcal{V}_{attribute}|$ denotes the number of possible values of the corresponding attribute. This becomes prohibitively expensive as we increase the number of attributes.

### A.2. Flow Matching Stability under its Closed-Form Formula

In addition to the assignment metric in Fig. 2, which is a discrete metric as it takes the argmax for the nearest neighbor training sample at the end of the trajectory, we also report a real-valued path deviation metric. Specifically, we generate trajectories by solving an ODE of the velocity obtained by the closed-form based on the full and pruned datasets, and calculate the path deviation along both trajectories: $\int \|x_t^{full} - x_t^{pr}\|_2 \, dt$. The metric is reported in Fig. 8. At $\mathrm{pr} = 0.8$, for example, the median is 0.58, with 95% of the samples having a deviation below 1.22, which is small compared to unrelated trajectories generated under different noise seeds using the full dataset ($80.15 \pm 2.07$ for CelebA-HQ, and $73.51 \pm 1.12$ for Synth, $mean \pm std$ over pairs).

To examine the behavior of the closed-form solution in the low-dimensional regime, we generate synthetic datasets by the same GMM and vary the dimensionality $D$. The softmax in eq. 2 tends to be less concentrated, such that multiple training samples can contribute to the velocity $\hat{u}^*(x, t)$, which results in more assignment changes at the endpoint of the trajectory. We plot the results in Fig. 9. As $D$ increases, we observe that the assignment changes less.

**Sample dominance from the closed-form perspective.**   To examine sample dominance and test whether there is a specific subset of special samples whose removal irreparably breaks the model, we perform this additional experiment: we sampled 100k pairs $(x_t, t)$, where $x_t = (1 - t)x_0 + tx_1$ and evaluated $u(x, t)$ using the closed-form solution in Eq. 2. We then counted how often each training sample was the dominant one in the solution (based on its weighted contribution to the softmax). We found that this dominance frequency is distributed broadly across the dataset rather than being concentrated on a small subset of training samples, as seen in Fig. 10. We interpret this as evidence against the existence of one special subset whose removal would by itself break transport stability. Rather, as long as we do not create holes in the data distribution (e.g., by removing entire clusters as in the gender experiment), the overall transport is largely preserved and stability holds. In particular, if enough representative samples from each region of the distribution are retained, other samples with similar transport geometry can take over. In the closed-form solution, this corresponds to another nearby sample becoming dominant after removal. A complementary analysis in Table 4 shows that reduced datasets preserving coverage across clusters lead only to a small increase in the error bound relative to the unpruned dataset (roughly 2–3%), whereas removing entire clusters (last row) leads to a much larger error bound (about 15%). This is consistent with the perceived-gender removal experiment

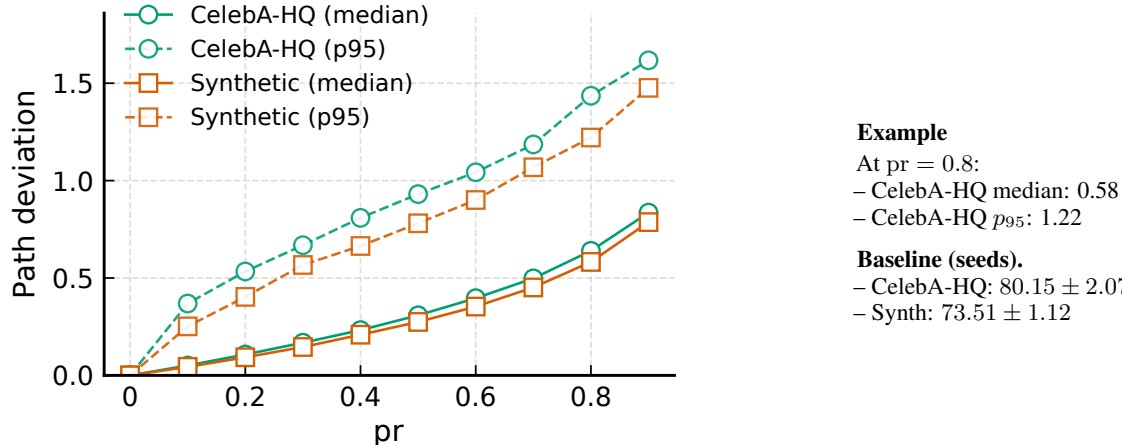

*Figure 8.* Path deviation vs. pruning fraction $pr$ (median and $p_{95}$).

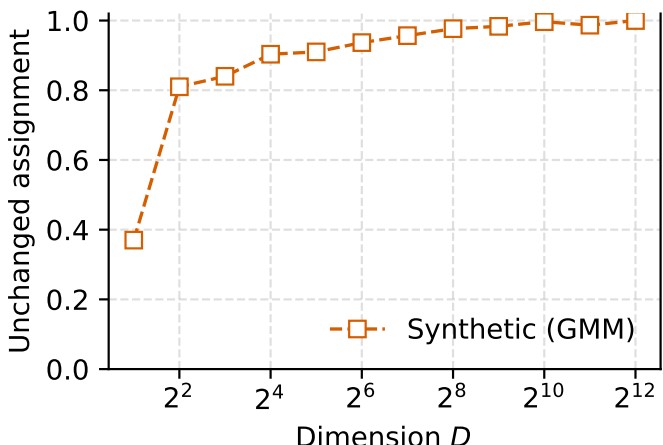

*Figure 9.* Unchanged assignment vs. synthetic data dimensionality under the flow-matching closed-form solution for $\mathrm{pr} = 0.8$.

in Fig. 4b and supports the view that coverage across the distribution matters more than preserving some specific subset.

### A.3. Stability

**Stability measurement.** We evaluate stability by comparing matched generations obtained from identical initial noise $x_0$ across independently trained models (different subsets, architectures, or training conditions), using embedding-space similarity and trajectory/assignment metrics.

**Detailed quantification.** In the manuscript, we reported a range of the similarity mean for each type of perturbation. In Table 3, we provide them in detail as $\mu \pm \sigma$ over $N = 4k$ generated pairs. In Table 3(a), all pruned variants maintain high similarity with *unpruned* (above 0.79), which is much higher than the similarity for unrelated (randomly shuffled) pairs ($0.37 \pm 0.12$). In Table 3(b), we compare models trained on disjoint subsets (a method and its inverse when $\mathrm{pr} = 0.5$), and they retain similarity too (above 0.72). In addition to perturbing the data distribution, changing the seed, indicating a different network initialization and sample shuffling preserve stability.

### A.4. Further stability tests

**Continuous KL-VAE** To test whether the observed stability stems from the discrete VQ-VAE latent representation, we repeat the CelebA-HQ pruning experiment using a continuous KL-VAE encoder-decoder that we trained on CelebA-HQ.

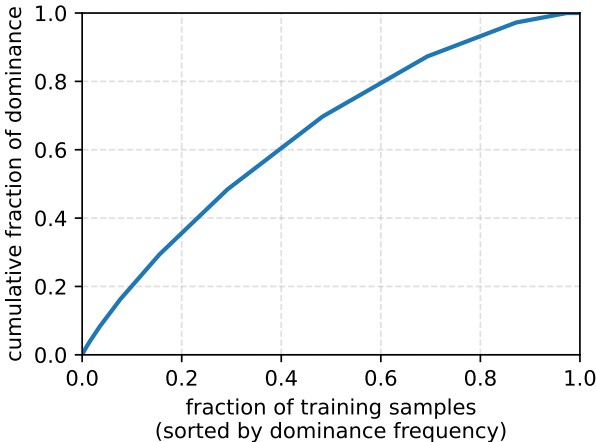

*Figure 10.* Cumulative dominance mass under the closed-form solution on CelebA-HQ latents. For example, the top 1% of samples account for only 2.7% of the mass, and covering 90% of the mass requires 74% of the samples.

| Random | $\mathcal{G}$ | $\mathcal{G}^{-1}$ | $\mathcal{L}$ | $\mathcal{L}^{-1}$ | $\mathcal{C}_p$ | $\mathcal{C}_p^{-1}$ | $\mathcal{C}_b$ | $\mathcal{C}_b^{-1}$ |
|---|---|---|---|---|---|---|---|---|
| $.83 \pm .11$ | $.79 \pm .12$ | $.80 \pm .12$ | $.80 \pm .12$ | $.80 \pm .13$ | $.80 \pm .12$ | $.81 \pm .12$ | $.81 \pm .12$ | $.79 \pm .13$ |

*(a)* ArcFace cosine similarity between each pruned model and *unpruned* for pr $= 0.5$. $N{=}4$k pairs matched by seed are evaluated. Unmatched pairs yield $0.37$.

| $\mathcal{G}^{1/-1}$ | $\mathcal{L}^{1/-1}$ | $\mathcal{C}_p^{1/-1}$ | $\mathcal{C}_b^{1/-1}$ |
|---|---|---|---|
| $.73 \pm .14$ | $.72 \pm .15$ | $.73 \pm .14$ | $.74 \pm .13$ |

*(b)* ArcFace cosine similarity between each method and its inverse for pr $= 0.5$, $N{=}4$k. Despite the training subsets being disjoint, the samples retain similarity.

*Table 3.* Stability quantification using ArcFace similarity ($\mu \pm \sigma$ over $N$ pairs) on CelebA-HQ under various perturbations.

Specifically, we train one DiT-S/2 model on the full dataset and one on a subset obtained by $\mathcal{C}_b$ at pr $= 0.5$. At 120k iterations, the pruned model yields FID $= 26.00$ compared to $27.68$ for the unpruned model, and the ArcFace similarity between them is $s = 0.77 \pm 0.13$. This result suggests that the stability effect is not tied to VQ quantization, but rather to FM transport in a fixed latent coordinate system. We visualise the results of both variants of KL-VAE for an identical seed in Fig. 11. As noted earlier in the stability tests, changing the latent coordinate system alters the mapping of noise to sample, such that the samples decoded by KL-VAE are different from the ones decoded by VQ-VAE under the same seed ($s = 0.3243 \pm 0.09$).

**Higher resolution.** We additionally verified that stability extends to higher image resolutions. Specifically, we trained the model on CelebA-HQ using image resolution $512 \times 512$, corresponding to latents of dimension $4 \times 64 \times 64$ rather than $4 \times 32 \times 32$ for resolution $256 \times 256$. Both models were trained for $120k$ iterations using batch size 256. We then evaluated two variants: the unpruned and pruned variant using $\mathcal{C}_b$ at pr $= 0.5$. We obtain FID $= 27.06$ for unpruned, and FID $= 24.99$ for pruned, and $s = 0.84$ between these two variants. The results suggest that the stability trend holds at higher resolutions as well.

**Effect of source-target coupling.** We additionally test whether coupling plays a role in the observed stability, specifically, the random source-target coupling used in the vanilla rectified-flow objective. We perform an experiment on CelebA-HQ in which we keep the linear interpolation path fixed but replace the random source–target coupling with minibatch optimal-transport (OT) coupling: Given a batch of target latents $\{x_1^j\}_{j=1}^B \sim p_1$, we sample an equal number of source samples $\{x_0^i\}_{i=1}^B \sim p_0$ and solve the assignment problem within-batch. The stability obtained is nearly on par with random coupling ($s = 0.80$ vs $s = 0.81$). Models trained using OT and random vanilla remain highly similar, especially in the unpruned setting ($s = 0.8703$). This result indicates that stability is not mainly driven by the specific random-pairing choice.

**Effect of $t$ sampling.** This is another perturbation whose aim is to test whether stability breaks if we discretize the sample interpolation timesteps when computing $x_t = (1 - t)x_0 + tx_1$ during training, i.e. we replace the original sampling scheme $t \sim \mathcal{U}(0, 1)$ with sampling from a predefined grid of $K$ timesteps. For the model trained using $K = 21$, we

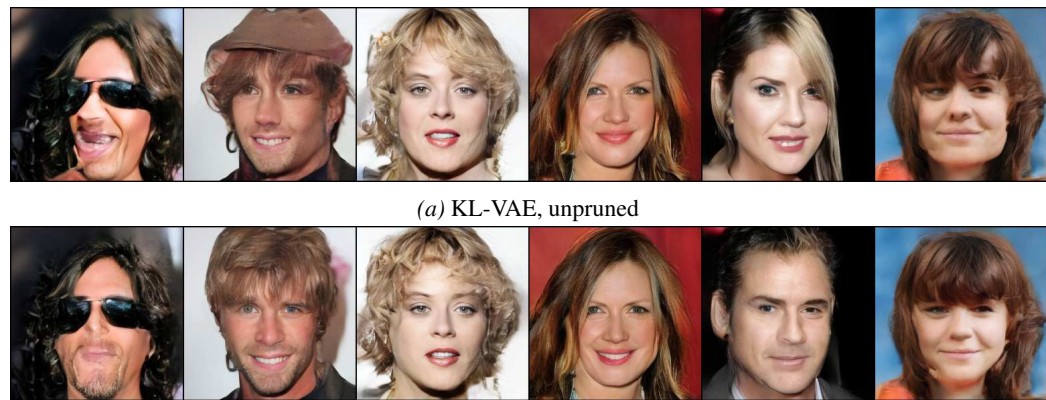

*(a)* KL-VAE, unpruned

*(b)* KL-VAE, $\mathcal{C}_b$, $\mathrm{pr} = 0.5$

*Figure 11.* Samples generated using an LFM model trained with a continuous KL-VAE. Columns correspond to identical initial noise seeds. The unpruned and $\mathcal{C}_b$-pruned models maintain high similarity.

find that it remains highly similar to the vanilla continuous-time sampling under matching seeds, with ArcFace similarity $s = 0.85 \pm 0.11$ and FID $= 25.6$ (vs FID $= 24.2$ for the vanilla model). With a sparser grid of $K = 11$, while the FID deteriorates substantially to FID $= 37.5$, it still retains a relatively high similarity with the vanilla model at $s = 0.79 \pm 0.12$.

### A.5. ImageNet

Since FID uses features extracted by an Inception network trained on ImageNet, which could introduce bias, we additionally report the Fréchet Distance using CLIP features ($\mathrm{FD}_{\mathrm{CLIP}}$). Figure 12 additionally shows precision, recall and F-score versus pr in both Inception and CLIP spaces. The results are largely consistent with FID, indicating that these results are not an artifact of the specific feature extractor. In addition, the fidelity and diversity are not diminished, as reflected in the precision/recall curves, indicating that there is no trade-off in performance.

### A.6. FFHQ

The top panel in Fig. 16 displays qualitative results on FFHQ, while the lower panel shows the FID curves vs pr. Qualitatively, we observe that stability holds and that there is no perceptible degradation in quality across the different variants. Quantitatively, up to $\mathrm{pr} = 0.5$, $\mathcal{C}_{b/p}^{-1}$ and *random* sampling perform nearly on par with *unpruned*. Compared with CelebA-HQ and ImageNet, $\mathcal{C}_{b/p}^{-1}$, which select less typical samples, outperform $\mathcal{C}_{b/p}$, suggesting that various pruning methods affect the generation distinctly based on the underlying distribution of the dataset.

To verify that stability holds on FFHQ as well, we compute ArcFace similarity. Across the pruning criteria, each pruned variant yields $0.81 \pm 0.12$ on pairs matched by seed with *unpruned* for $\mathrm{pr} = 0.25$ (unmatched pairs yield $0.3 \pm 0.1$). For $\mathrm{pr} = 0.5$, the similarity is $0.77 \pm 0.12$. For a criterion and its inverse, the similarity is $0.71 \pm 0.13$.

### A.7. CelebA-HQ

We also report precision, recall and F-score on CelebA-HQ in Fig. 17 for a more comprehensive generative evaluation. We include the FID curves again for a direct comparison. Under pruning, CelebA-HQ introduces a slight trade-off between precision and recall, which we cannot infer from FID alone. This is in contrast to ImageNet, which maintained performance across all metrics under a limited compute budget, indicating the behavior depends on the dataset.

### A.8. Coarse-to-Fine (C2F)

We visualize examples of this two-model approach in Fig. 18. When only the DiT-S/2 small architecture is used (row 1), we observe that the images have artifacts reflected in occasional blotches. When DiT-XL/2, a larger capacity model, is used (row 2), these artifacts disappear and the images appear sharper. In the *coarse-to-fine* approach, we observe that the fine model corrects the artifacts of the weaker coarse model, and matches the performance of the fine model while saving substantially in inference time.

In order to emphasize the importance of stability in our proposed *coarse-to-fine* approach, we stitch the fine model with a coarse model trained using only male-perceived gender ($\text{C2F}_{\text{male}}$), since this is one of the perturbations that partially broke stability. We observe that some samples that would have been generated as female in the original and stable models, have artifacts, as Coarse's trajectory is generating a male person, while Fine's trajectory is generating a female, resulting in discontinuity at the seam, despite it working for some samples such as the first column. This result is consistent with a drop in the FID metric as well, which we reported earlier in the manuscript to be 44.92.

## B. Theoretical Error Bounds.

Through the closed-form analysis and its link to the training objective of an LFM network, we provided insight into why stability holds in LFM across different subsets of the data. Here, we perform a complementary analysis using the error bounds established in (Benton et al., 2023).

**Error bounds**  Benton et al. (2023) have devised the following inequality on the Wasserstein distance between the true probability flow (PF) $\pi_1$ and a learned PF $\hat{\pi}_1$, specifically,

$$
W_2(\hat{\pi}_1, \pi_1) \leq \exp\left(\int_0^1 L_t \, dt\right) \underbrace{\left(\int_0^1 \mathbb{E}_{x \sim \pi_t}\big[\|v_\theta(x,t) - v^\star(x,t)\|_2^2\big] \, dt\right)^{1/2}}_{\epsilon} \tag{5}
$$

where $v^\star(\cdot, t)$ denotes the ground-truth velocity field, $\pi_1$ its corresponding probability flow, and $\hat{\pi}_1$ is the distribution induced by $v_\theta(x_t, t)$.

**Error bound results on CelebA-HQ.**  We compute these bounds by approximating the Lipschitz term and expected velocity error over a time grid $t \in (0, 1)$. We use an LFM model trained on CelebA-HQ (complete dataset) to approximate the Lipschitz constant, specifically, we compute the median spectral norm $\nabla_x v_\theta(x, t)$ via power iteration with Jacobian–vector products (JVP) (Behrmann et al., 2019) and obtain $\exp\left(\int_0^1 L_t dt\right) = 24.29$.

The velocity field $v_\theta(x, t)$ is computed using the respective model that we want to bound. Additionally, we use CelebA-HQ training set to approximate $v^*(x, t)$ by the closed-form formula in eq. 2, where we draw samples $x_t = (1 - t)x_0 + tx_1$ based on $x_1$ from the validation set.

We report the resulting $W_2$ bounds in Table 4. The bounds are so close to each other and do not correlate accurately with the FID, i.e. a lower bound does not indicate a better FID, hence cannot be used to deduce performance or stability. The most noticeable increase in error is incurred when we remove half the label-agnostic clusters (analogous to the gender removal experiment), which we have shown to break stability. This larger error bound could signal when stability is violated rather than when it is maintained. Since the Lipschitz constant is a global estimate and is identical across all variants, we hypothesize that the increase happens because dropping clusters creates discontinuities in the distribution, which forces the velocity field to increase in order to overcome these forming low-density regions.

In all model variants that we examined, the bounds were larger than the small generation distributional difference that we obtained through a reduced dataset. These large bounds stem from the bound's dependency on the Lipschitz constant in the exponent based on Grönwall's inequality, and are often too large to be practical (Benton et al., 2023). Furthermore, this bound applies to a single model when compared with the optimal velocity, such that comparing two models based on their upper bound value is meaningless. For instance, the actual error of one model could be well below the obtained bound despite having a larger upper bound than the other model.

Nevertheless, if we want to compare models, the Wasserstein distance satisfies the triangle inequality, which extends the bound to include comparison between the various model variants. Specifically, if we want to compare the $W_2$ bounds between two models $i$ and $j$, the triangle inequality for $W_2$ metric satisfies

$$
W_2\big(\hat{\pi}_1^i, \hat{\pi}_1^j\big) \leq W_2\big(\hat{\pi}_1^i, \pi_1\big) + W_2\big(\pi_1, \hat{\pi}_1^j\big). \tag{6}
$$

Comparing two models therefore would entail summing their $W_2$ theoretical bounds with respect to the true distribution. For instance, comparing $\mathcal{C}_b$ and $\mathcal{G}$, we obtain $W_2\big(\hat{\pi}_1^{\mathcal{C}_b}, \hat{\pi}_1^{\mathcal{G}}\big) \leq 1.51e3 + 1.51e3$. Such a large bound is hardly indicative of stability across models, but we have observed that these models have a high similarity between their generated images.

| Model variant | Theoretic bound $W_2$ |
|---|---|
| *Unpruned* | 1.47e3 |
| $\mathcal{L}$ | 1.51e3 |
| $\mathcal{L}^{-1}$ | 1.52e3 |
| $\mathcal{C}_b$ | 1.51e3 |
| $\mathcal{C}_b^{-1}$ | 1.50e3 |
| $\mathcal{G}$ | 1.51e3 |
| $\mathcal{G}^{-1}$ | 1.51e3 |
| Half clusters removed | 1.70e3 |

*Table 4.* FM $W_2$ theoretical error bounds on various models.

## C. Experimental setup

### C.1. Training

**DiT.**  We use the same training hyperparameters prescribed in DiT (Peebles & Xie, 2023). For each dataset, we precompute the image latents using the corresponding trained VQ-VAE offline to save the repeated VAE forward passes during training. We train each model variant for $200k$ iterations on CelebA-HQ and report the best checkpoint, which occurs at approximately $140k$ iterations, $200k$ on ImageNet, and $150k$ on FFHQ, respectively, using a global batch size of 128 on four NVIDIA A100 GPUs. Training on ImageNet using DiT-XL/2 architecture requires 48 hours of 4 GPUs, while training DiT-S/2 on CelebA-HQ requires $\approx 14$ hours, and DiT-B/2 on FFHQ $\approx 13$ hours. On CelebA-HQ, we report the best FID in this $200k$ budget, which occurs at $\approx 140k$ and plateaus thereafter.

**VQ-VAE.**  We train a VQ-VAE on each of the target datasets separately, in order to prevent distributional interference from other datasets when evaluating the generated images. For example, in our initial setup, when we used a pretrained VAE, we observed that when training on a very small number of images, the model was replicating celebrity faces not present in CelebA-HQ, which made it difficult to detect overfitting by comparing with our training set only. Accordingly, we trained three VQ-VAEs, one for CelebA-HQ, FFHQ, and ImageNet datasets each. We use Adam optimizer with $\beta_1 = 0.9$, $\beta_2 = 0.999$, $\epsilon = 10^{-8}$ with a fixed learning rate (LR) $10^{-4}$. Additionally, following VQ-GAN (Esser et al., 2021), we introduce a discriminator after $5k$ iterations and add adversarial and perceptual loss terms, gradient clipping at 2.0 and EMA decay of 0.999 of generator weights. The global batch size is 64 on 4 GPUs, and we train for $100k$ iterations with image resolution $256 \times 256$.

**Evaluation metrics.**  In addition to FID, we include *precision*, *recall* and *F-score* (Kynkäänniemi et al., 2019). These metrics do not refer to classification metrics, but rather to evaluations in the feature space of the real and generated distributions. Specifically, precision measures image fidelity, recall measures coverage while F-score combines both by their harmonic average.

## D. Notation Lexicon

**Acronyms.**

- **FM**: flow matching.

- **LFM**: latent flow matching (FM in a learned latent space, obtained by our trained VQ-VAE or KL-VAE).

- **ODE/SDE**: ordinary/stochastic differential equation.

- **EMA**: exponential moving average.

- **C2F**: the proposed Coarse-to-Fine approach.

- **FID**: Fréchet Inception Distance using Inception network.

- **FD$_{\text{clip}}$**: Fréchet Distance in CLIP space.

- **GMM**: Gaussian Mixture Model.

**Core notation.**

- $S$: Training set
- $S'$: A training subset ($S' \subset S$)
- $pr \in [0, 1]$: *pruning fraction* (fraction removed), so $|S'| = (1 - pr)|S|$ and $\mathrm{pr} = 0$ corresponds to *unpruned*.
- $t_0$: seam point for coarse-to-fine trajectory splitting.
- $x_t$: state at time $t$; $x_0 \sim p_0$ (noise prior), $x_1 \sim p_1$ (data/latent distribution).
- $v_\theta(x, t)$: learned velocity field; $u(\cdot)$: target/ground-truth velocity in the FM objective.

**Pruning criteria**

- $\mathcal{G}$: gradient-based pruning.
- $\mathcal{L}$: loss-based pruning.
- $\mathcal{C}$: clustering-based pruning.
- $\mathcal{C}_b$: *balanced* clustering/pruning (equalizes cluster representation by pruning overrepresented clusters).
- $\mathcal{C}_p$: *proportional* clustering/pruning (preserves the original cluster proportions).
- Superscript $^{-1}$: selects samples furthest from the cluster center for clustering; lowest-gradient/loss samples for gradient/loss-based selection criteria respectively. No superscript or superscript $^{+1}$ denotes selecting samples nearest to the center; highest-gradient/loss samples.
- $\mathcal{C}_b^\kappa$: kernel-based selection within each cluster.

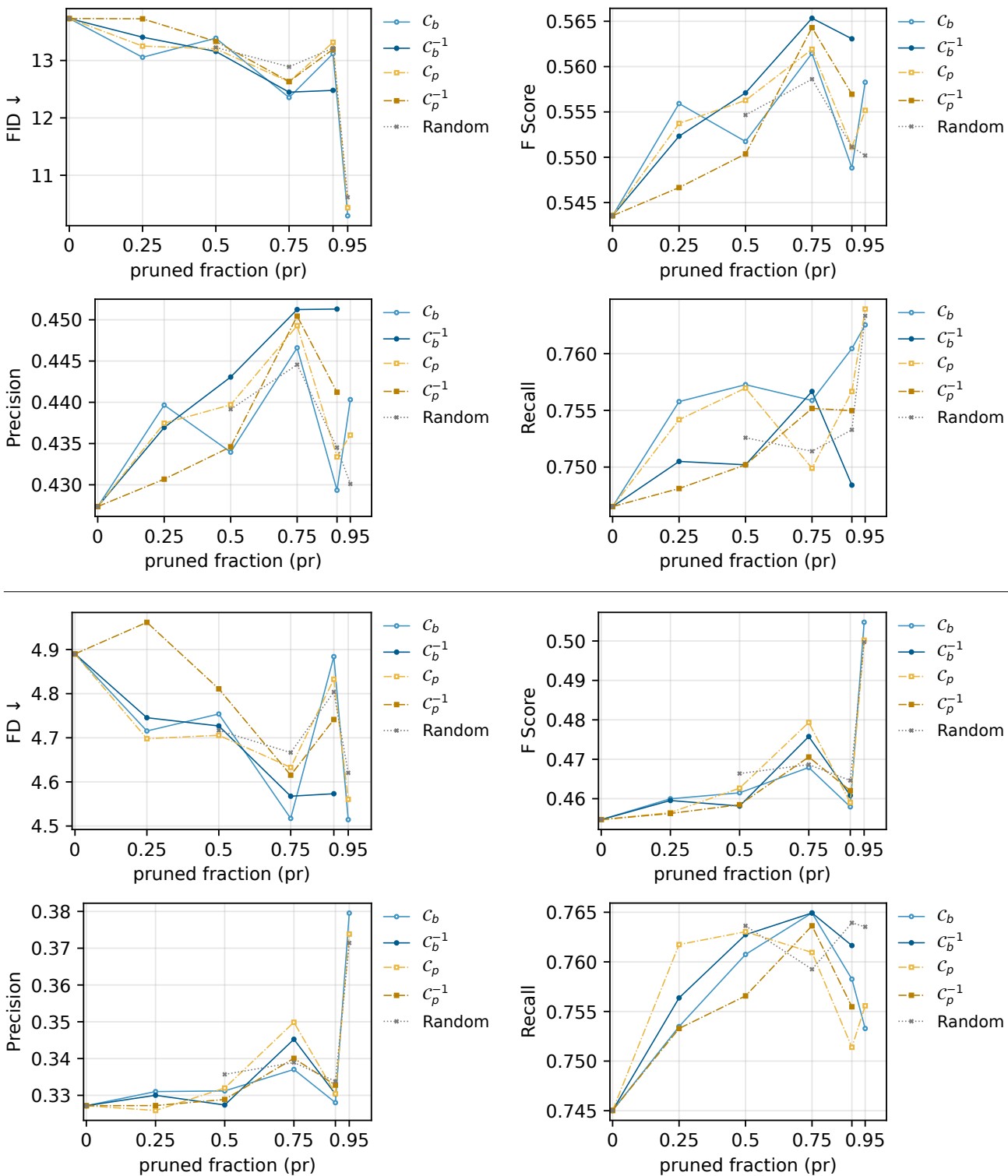

*Figure 12.* ImageNet evaluation metrics vs pruning fraction at a fixed budget of $200k$ iterations. **Top:** Inception-based metrics (FID, F-score, precision, recall). **Bottom:** Analogously, CLIP-based metrics.

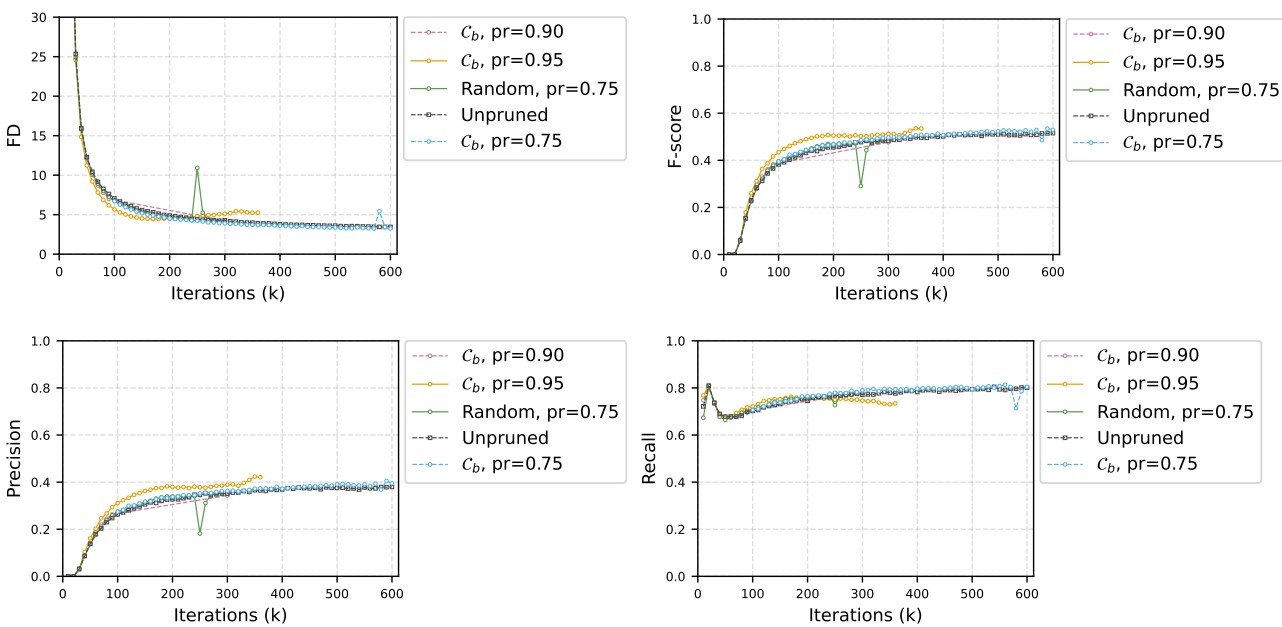

*Figure 13.* CLIP-based evaluation metrics on ImageNet vs iterations: $\text{FD}_{\text{CLIP}}$ (top-left), F-score (top-right), precision (bottom-left), and recall (bottom-right).

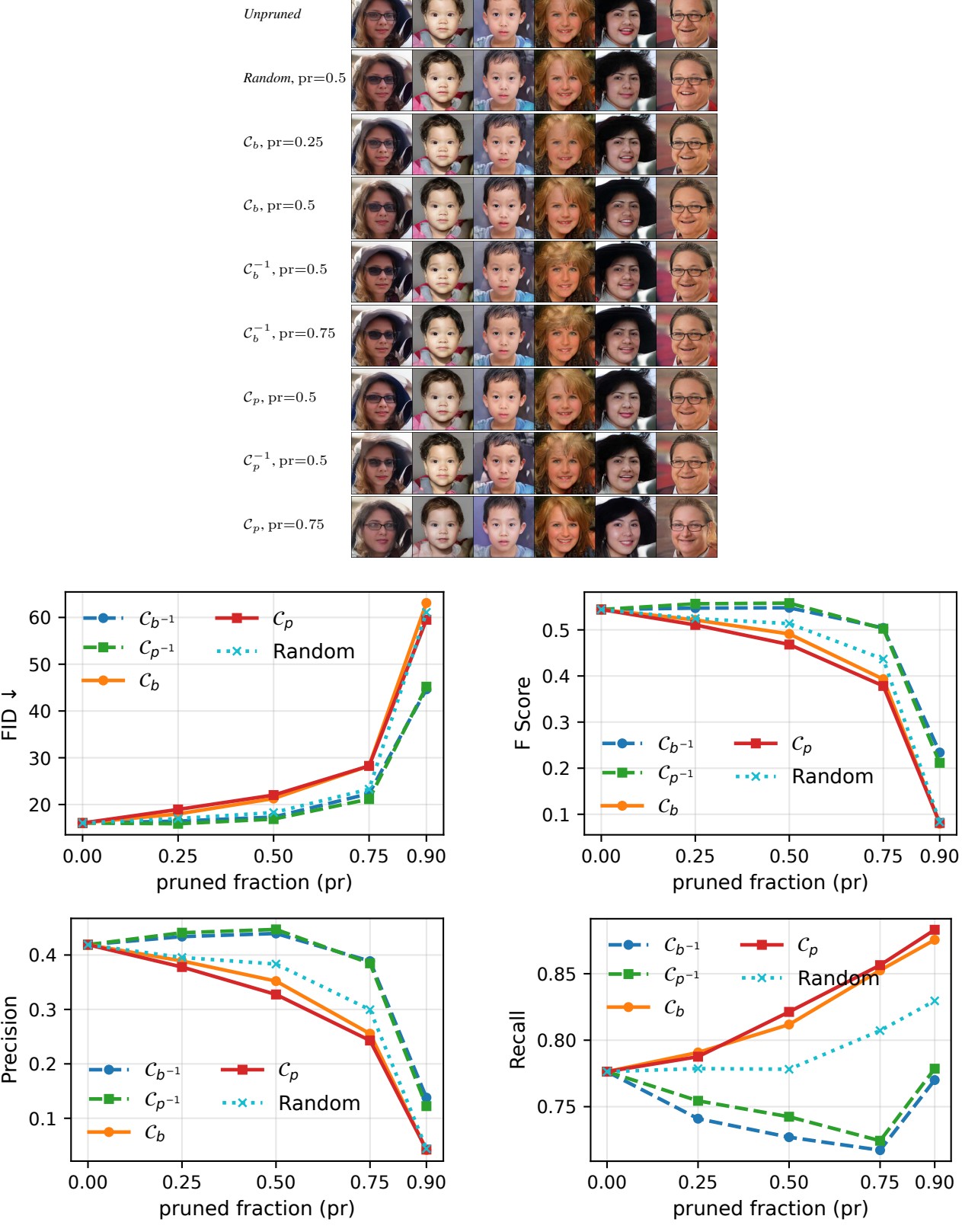

*Figure 16.* **TOP**: qualitative FFHQ samples under various pruning methods and fractions. For example, the subsets used in row 4 and 5 are disjoint. **Bottom**: FFHQ evaluation metrics vs pruning fraction pr at a fixed budget of $150k$ iterations.

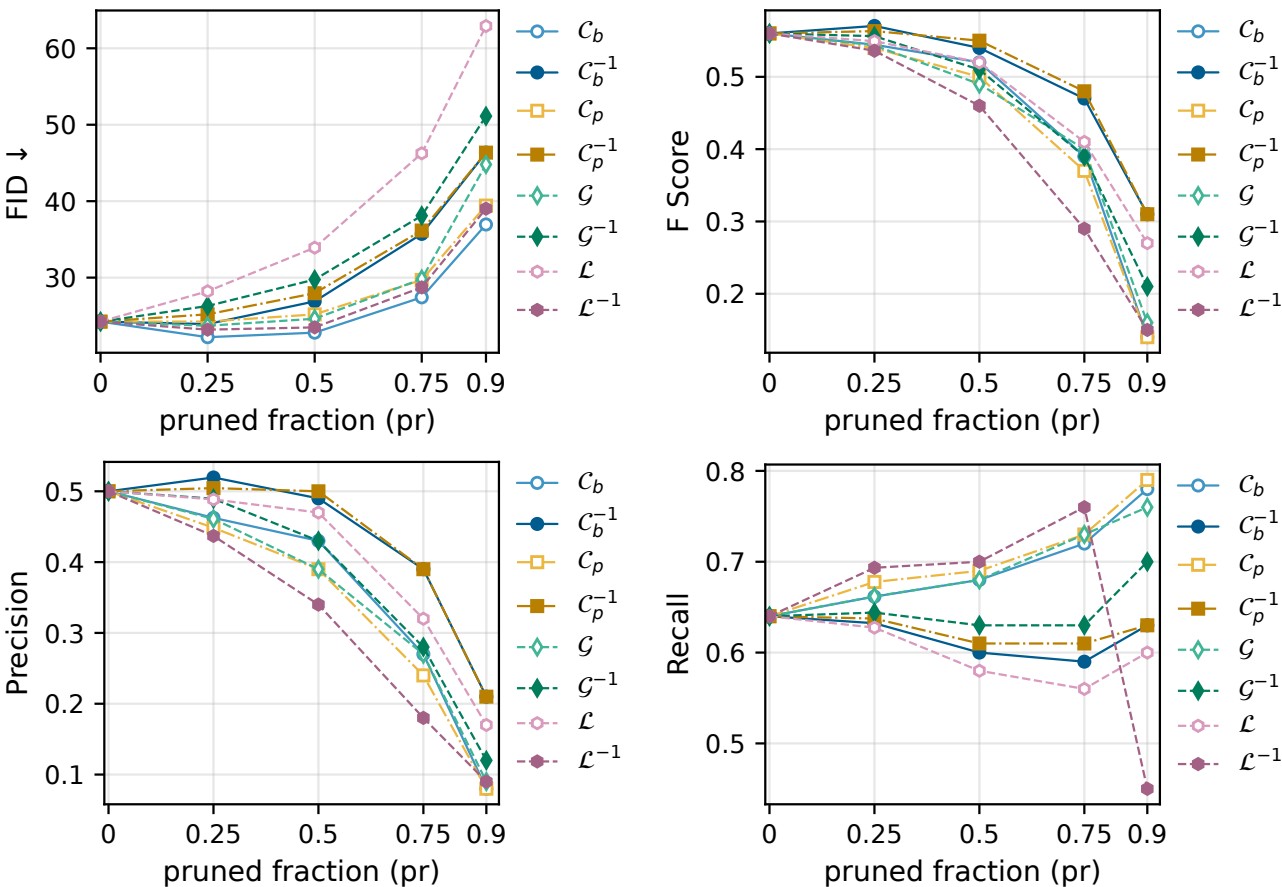

*Figure 17.* CelebA-HQ evaluation metrics vs pruning fraction pr at a fixed budget: FID (top-left), F-score (top-right), precision (bottom-left), and recall (bottom-right).

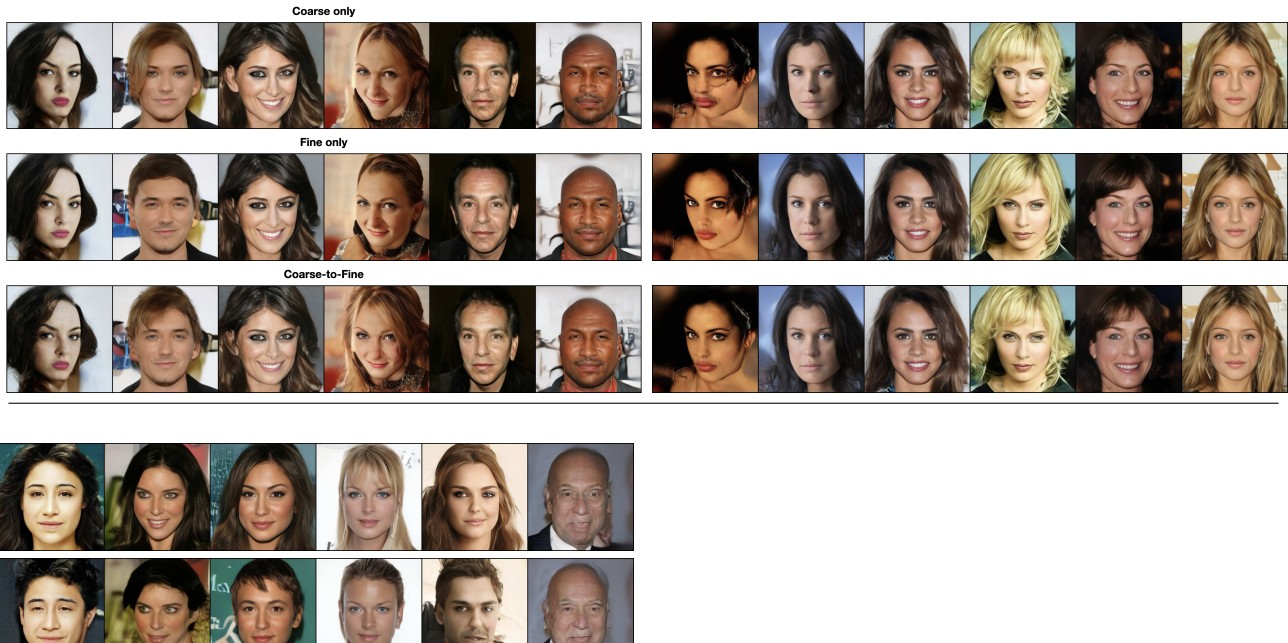

*Figure 18.* Qualitative results for the two-stage *coarse-to-fine* approach. Top: two different initial noise vectors (columns), shown across coarse-only, fine-only, and coarse-to-fine. Bottom: coarse-to-fine when stability holds (top row) versus the $C2F_{male}$ experiment that partially breaks stability (bottom row).

