# OpenReview forum: "Exploring and Exploiting Stability in Latent Flow Matching"
_ICML.cc/2026/Conference — ICML 2026 regular_

### Official Review · Reviewer_jnXf · 2026-03-03

**Soundness:** 3
**Presentation:** 3
**Significance:** 3
**Originality:** 3
**Overall Recommendation:** 4
**Confidence:** 3

**Summary:**

The authors found that flow matching models are quite robust to data or model changes. Particularly, the trajectory starting from the same source point may move to similar end points with the learned vector field, even when training data are significantly pruned or model sizes/architecture is altered. This finding motivates authors to propose a coarse-to-fine two-stage model, where the first coarse model learns the trajectory between source point and an intermediate point, and the fine model continues to learn the trajectory to the end. According to the stability finding of LFM, the authors choose to use much fewer data and smaller model to train the coarse part, so that the whole training and inference can be accelerated. Experimental results demonstrate the effectiveness of proposed method.

**Compliance With Llm Reviewing Policy:**

Affirmed.

**Key Questions For Authors:**

1. I don't quite understand the FM stability. I know it is illustrated by many experimental results. Can the authors explain why is this possible? I noticed on page 2 that "...indicating that one training sample dominates." I don't understand why there is one training sample will dominate in early stages and how it is linked to FM stability. Is this a universal phenomenon, if so, why?
2. Which part in the proposed method will make sure you don't prune this important sample when you do the pruning?

**Limitations:**

yes

**Strengths And Weaknesses:**

## Strengths
1. The stability of LFM is a novel and interesting finding, and it has potential in speeding up the training and inference of LFM.
2. Visualization in this paper is very nice, e.g. fig 1, 2, including both synthetic toy dataset and real dataset. The motivation and idea is easy to get.
3. Experimental design is quite complete. The authors consider three scores, and each is with a nearest and furthest samples constructed variant. Fig 4 studies different perturbations, and Fig 5 ablates the choice of t_0.


## Weaknesses
1. Although the comparison among many variants of the proposed method is well illustrated, I don't see a straightforward comparison with SOTA LFM/FM models. Particularly, since the major advantage of proposed method is lower training and inference cost, readers may wonder how much training/inference cost is reduced compared to SOTA methods, and how much performance is sacrificed.

---

> ### Author Rebuttal · Authors · 2026-03-28
>
> We thank the reviewer for thinking our finding is interesting and novel, and for raising these insightful questions, which we hope to address below.
>
> (i) **Comparison with SOTA**. Stability is a property of LFM rather than the specific architecture. Therefore, applying data pruning or Coarse-to-Fine is not tied to a particular backbone. Our choice of the DiT model with latent Flow Matching was motivated by it being a strong and well-established transformer-based architecture. We agree that comparisons with additional recent methods would further strengthen the empirical evaluation. However, our main contribution is not achieving a new SOTA, but rather demonstrating the stability phenomenon and exploiting it in two practical ways using a fixed backbone while varying its capacity.
>
> (ii) **One-sample dominance**. This is indeed a surprising phenomenon. It was formally analyzed by Bertrand et al., who show that in the closed-form solution of flow matching, the contribution to the vector field can be dominated by a small subset of nearby samples. Since the learned model is trained via the FM objective (eq. 1) to approximate this closed-form solution, it can inherit similar robustness properties. Intuitively, this suggests that the learned transport is governed by local structures in the data distribution, which makes it stable even when many samples are removed. This also connects to (iii): as long as the retained dataset adequately represents the underlying distribution, other nearby samples can take over the role of removed ones. We will clarify this connection in the manuscript.
>
> (iii) **Are important samples pruned?** Thanks for raising this important point. Our analysis of the deviation between flows trained on the pruned and full datasets indicates that there is no single subset of samples whose removal irreparably breaks the model. Instead, as long as we do not create holes in the data distribution (e.g., by removing entire clusters as in the gender experiment), the overall transport is largely preserved and stability holds. In particular, as long as enough representative samples from each region of the distribution are retained, other samples with similar transport geometry can take over. In the closed-form solution, this corresponds to another nearby sample becoming dominant after removal. We will include this analysis in the appendix for more clarification.

---

> > ### Author Rebuttal · Reviewer_jnXf · 2026-04-03
> >
> > Thanks for the response.
> >
> > (i) I think I made it clear in my initial review that, by comparing with SOTA, I don't mean you need to beat them on FID, but on **training and inference cost**, because the proposed method is more data- and parameter-efficient. How much improvement you can make on speed, and how much FID gap you will sacrifice. The comparison would be very much like DDIM vs DDPM, where DDIM may not necessarily beat DDPM on FID, but it is way faster.
> >
> > (ii) Resolved.
> >
> > (iii) I don't see the details of "our analysis" so I am not sure if it makes sense. "as long as we do not create holes in the data distribution..." and "as long as enough representative samples from each region of the distribution are retained..." are conditions rather than guarantees.

---

> > > ### Author Response · Authors · 2026-04-04
> > >
> > > Thank you for the clarification and the follow-up. We would like to clarify these points further.
> > >
> > > (i) **Comparison with SOTA in terms of speed/cost**.
> > > We understand the point better now. Our claim is not that the method improves SOTA FID, but that it can reduce training or inference cost while preserving competitive quality. For inference, our Coarse-to-Fine approach yields a measured $2.15\times$ speedup while maintaining comparable output quality, as reported in the paper. For training, the pruning results show that under a fixed compute budget, reduced datasets can reach similar quality faster, with the main gain being improved optimization efficiency rather than a strictly better optimum. We will revise the manuscript to make this cost-versus-quality tradeoff more explicit.
> > >
> > > (ii) **On whether there is a special subset of important samples**.
> > > We agree that our earlier wording described conditions rather than guarantees, and thank you for pointing this out. What we can support empirically is the following:
> > >
> > >    (1) Our appendix analysis in Table 4 shows that reduced datasets preserving coverage across clusters lead only to a small increase in error relative to the unpruned dataset (roughly 2–3%), whereas removing entire clusters (last row) leads to a much larger degradation (about 15%). This is consistent with the gender-removal experiment in Fig. 4(b), and supports the view that coverage across the distribution matters more than preserving some specific subset.
> > >
> > >    (2) To test whether there exists a particularly special subset of constant dominating samples, we sampled 100k pairs $(x_t,t)$, where $x_t=(1-t)x_0+tx_1$ with $x_0\sim\mathcal{N}$, and evaluated u(x,t) using the closed-form solution in Eq. 2. We then counted how often each training sample was the dominant one in the solution (based on its weighted contribution to the softmax). We found that this dominance frequency is distributed broadly across the dataset rather than concentrated on a small subset of training samples. We interpret this as evidence against the existence of one special subset whose removal would by itself break transport stability.
> > >
> > > Taken together, these results do not provide a formal guarantee, but they do support the empirical conclusion that preserving representative coverage is more important than retaining any particular fixed subset of samples. We will include this analysis more explicitly in the appendix and clarify the wording in the manuscript.

---

### Official Review · Reviewer_XLX6 · 2026-03-12

**Soundness:** 3
**Presentation:** 3
**Significance:** 2
**Originality:** 2
**Overall Recommendation:** 3
**Confidence:** 3

**Summary:**

This paper investigates the stability properties of Latent Flow-Matching (LFM) models, showing that they are robust to perturbations including dataset reduction and model capacity shrinkage. The authors characterize this stability through the tendency to generate similar outputs under identical noise seeds and relate it to flow matching theory. They exploit this stability for practical applications: training on reduced datasets for faster convergence and reduced annotation effort, and a two-model coarse-to-fine approach using lightweight architecture for early timesteps and higher capacity for later timesteps. The paper introduces three sample-scoring criteria to identify informative samples and demonstrates results on multiple datasets, claiming data savings and more than two-fold inference speedup with comparable output quality.

**Compliance With Llm Reviewing Policy:**

Affirmed.

**Key Questions For Authors:**

Regarding two-stage sampling, if we want to use the kit approach in a production environment, do we need to readjust the switching point for different datasets, or can we directly use the switching strategy found in the paper?

**Limitations:**

he paper primarily discusses the benefits of stability, but it doesn't adequately address some issues encountered in practical deployments.

**Strengths And Weaknesses:**

Strengths:

1. The stability observation is interesting and practically useful - training on reduced data without quality degradation could save compute and annotation effort.

2. The two-stage coarse-to-fine approach is a sensible way to speed up inference, and the reported 2x speedup is promising for deployment.

3. Experiments cover multiple datasets showing the method works across different settings.

Weaknesses:

1. The claim that performance doesn't degrade with significantly reduced data is surprising and lacks comparison with standard data pruning baselines like core-set selection.

2. The two-model approach adds deployment complexity (memory overhead, maintenance) that isn't quantified or compared against simpler acceleration methods.

3. Evaluation is limited to standard image datasets; unclear if stability holds for higher resolution or video generation.

---

> ### Author Rebuttal · Authors · 2026-03-28
>
> We thank the reviewer for their review and critique of the paper. We address these questions and concerns below.
>
> (i) **Performance maintained despite significant data reduction**. We refer to answer (i) in our response to reviewer CYev. Regarding standard pruning strategies, the employed pruning strategies based on loss/gradient/clustering are standard pruning methods in discriminative models which we adapted to FM models. Core-set is also an important baseline, we therefore conducted an additional experiment using pr=0.5, and obtained FID 26.25, which is worse than the methods used in the paper (cf. the FID table in Fig. 3). This is consistent with the behavior of $C_b^{-1}$, which also prioritizes distant samples and shows degraded performance (FID 26.77 vs 22.8 for $C_b$). We will include this experiment and its analysis in the revised manuscript.
>
> (ii) **Deployment complexity**. The memory overhead pertains to loading and holding the smaller model in memory, which is much smaller in size than the full-capacity model. There is a slight trade-off between handling two models and inference time, and we will make this trade-off explicit in the manuscript.
>
> (iii) **Stability for higher resolution images and videos**. We additionally verified that stability extends to higher image resolutions. Specifically, we trained a flow-matching model on CelebA-HQ using image resolution 512x512, corresponding to latents of dimension 4x64x64 rather than 4x32x32 for resolution 256x256. Both models were trained for 120k iterations using batch size 256. We then evaluated two variants: the unpruned variant, and $C_b$ using pruning fraction pr=0.5. We obtain FID=27.06 for unpruned, and FID=24.99 for  $C_b,pr=0.5$, and arcface similarity metric between the models s=0.84 (for reference, it was s=0.81 for the same variants at resolution 256). These results suggest that the stability trend holds at higher resolutions as well.
>
>  We did not experiment with videos since it is a different modality and requires different adaptations of the pruning methods. We leave this as future work. However, we had experimented with random pruning on a temporal human motion dataset (HumanML3D [\*]) in a conditional setup with text. In this experiment, we removed entire sequences from the dataset for various pruning fractions during the model training, and observed a similar phenomenon at inference: the generated sequences conditioned on text remained similar and FID remained stable up to pr=0.75:
>
> pr  |   0   |  0.25 |  0.5  |  0.75 |  0.9  |
>
> -------------------------------------------------------
>
> FID | 0.549 | 0.373 | 0.435 | 0.697 | 0.715 |
>
> We will include these experiments in the appendix.
>
> (iv) **KiT approach for C2F**. Our results (cf. Fig. 5) indicate the approach is not sensitive to the choice of $t_0$ up to 0.5. It is true that our method does not find this point adaptively, which we leave as future work.
>
> \* Generating Diverse and Natural 3D Human Motions From Text, Guo et al.

---

> > ### Author Rebuttal · Reviewer_XLX6 · 2026-04-03
> >
> > I thank the authors for their detailed response. The additional experiments on higher-resolution images (CelebA-HQ 512×512) and the HumanML3D motion dataset (iii) are appreciated and provide reasonable evidence that stability generalizes beyond the original settings. The core-set baseline comparison (i) is also a helpful addition. However, two concerns remain partially unresolved. First, regarding deployment complexity (ii), the authors acknowledge the memory trade-off but do not provide quantitative comparisons against simpler acceleration methods (e.g., reducing NFE, model distillation), which would be important for justifying the practical value of the two-model approach. Second, regarding the switching point (iv), the claim that performance is insensitive to $t_0$ up to 0.5 is only empirically observed on a limited set of datasets, with no theoretical justification for why this should hold in general. Leaving adaptive selection entirely to future work means that practitioners would need to manually tune this hyperparameter for each new dataset, which undermines the claimed practical benefits of the method. Based on the above, I choose to maintain my original score.

---

> > > ### Author Response · Authors · 2026-04-04
> > >
> > > Thank you for the follow-up and for taking the additional experiments into account.
> > >
> > > (i) **Comparison with other acceleration baselines**. We agree that quantitative comparison to simpler acceleration baselines would strengthen the practical discussion. Our main point here is that the proposed speedup arises from the stability property of LFM itself, rather than from reducing NFEs, distillation, or modifying the FM objective. In that sense, our approach is complementary rather than directly competing with these methods, and could in principle be combined with them. We will make this positioning clearer in the revised manuscript.
> > >
> > > (ii) **The switching point $t_0$**. We agree that $t_0$ is currently a hyperparameter of the method rather than something chosen adaptively. Empirically, our results suggest that performance is not highly sensitive to this choice up to $t_0=0.5$ on the datasets that we evaluated. In practice, selecting $t_0$ is inexpensive, since it can be tuned on a small validation set. We agree that an adaptive criterion would further improve usability, and we will highlight this more clearly as a limitation and direction for future work.

---

### Official Review · Reviewer_CYev · 2026-03-12

**Soundness:** 3
**Presentation:** 3
**Significance:** 3
**Originality:** 3
**Overall Recommendation:** 4
**Confidence:** 3

**Summary:**

This paper studies the stability of latent flow matching, where samples generated from the same initial noise produce similar samples after substantial dataset pruning or changes in model capacity. Based on this observation, the paper connects the phenomenon to the closed-form structure of flow matching and uses it to motivate both dataset pruning and a coarse-to-fine model approach. Experiments suggest that this stability can be used to improve training and inference efficiency.

**Compliance With Llm Reviewing Policy:**

Affirmed.

**Final Justification:**

The paper presents an interesting empirical finding with reasonable originality, and the rebuttal addressed most of my main concerns. While some interpretations still feel somewhat stronger than evidence, the overall contribution is solid enough. So, I keep my score as weak accept.

**Key Questions For Authors:**

1. The paper shows several cases where pruning improves FID, sometimes even over the unpruned model. (Figure 3, Figure 5, ImageNet). Could the authors clarify how they interpret this? I had the impression that comparing models at the same number of iterations may not be entirely fair, since the unpruned model could still be undertrained.

2. The paper motivates pruning through the dominant-sample intuition, but random pruning also seems to work surprisingly well, and on ImageNet it even improves the unpruned model. It was not fully clear to me how important the specific pruning rule is in practice. I would appreciate more discussion of this point.

3. The latent perturbation results are interesting, but I was not fully sure how to interpret them. Could the authors clarify how much the observed drop is due to a real change in transport behavior, as opposed to degradation in the latent representation or decoding quality itself? Additional analysis along this line would make this claim more convincing.

**Limitations:**

The paper includes an impact statement. However, the limitations of the method are not discussed. A brief discussion of these limitations would strengthen the paper.

**Strengths And Weaknesses:**

**Soundness**
* This paper is solid overall, and the main empirical finding is supported by a broad set of experiments.
* The closed-form analysis is helpful for intuition, and the coarse-to-fine results are reasonably convincing.
* However, some claims are stronger than the evidence, the source of pruning gain, and the interpretation of the latent perturbation results.
* And, it is also somewhat unclear whether the reported pruning gains reflect a better model or simply faster optimization under a fixed iteration budget.

**Presentation**
* The paper is clearly written and easy to follow.
* The main idea is clear, and the figures help make the qualitative behavior easy to understand.
* However, some of the more surprising results are not fully explained, especially why pruning sometimes improves FID and how the latent perturbation results should be interpreted.
* And, some of the quantitative results in the experimental section are described mostly in text, which makes it harder for me to compare.

**Significance**
* I think the paper studies a meaningful and relevant question.
* The stability observation is interesting, and the possible effects for data efficiency and faster inference are useful.
* Also, the paper evaluates the idea across several datasets and settings, which helps support its broader potential beyond a single benchmark.

**Originality**
* The paper is reasonably original overall.
* The main novelty is in identifying this stability property of latent flow matching and using it to motivate pruning and coarse-to-fine generation.

---

> ### Author Rebuttal · Authors · 2026-03-28
>
> We thank the reviewer for recognizing the significance and originality of the paper, and for raising these important questions.
>
> (i) **The source of pruning gain and what it reflects**. On ImageNet, our claim in the paper is that the gain occurs under a limited training compute budget. For an identical number of training iterations, on a reduced dataset, the number of effective epochs is higher than on the full dataset. That is, the number of times each sample is seen by the model is higher. In this case, the model needs to fit itself to a smaller number of samples, which allows the model to fit the distribution faster and leads to faster convergence. Close to convergence, as seen in the FID vs iterations curve in Fig. 6, the gap between the best performing method and unpruned is narrowed down considerably. This suggests that the primary benefit is improved optimization efficiency rather than a strictly better optimum.
>
> (ii) **Interpretation of the latent perturbation**. Thanks for raising this point. Our motivation for applying an invertible transform to the latent was testing the hypothesis whether latent FM is learning some canonical transport independent of everything, including the latent space, as long as the dataset has enough representative samples of the distribution. If the model can recover its endpoint latent when inverting it back, this would support this hypothesis. Since this was not the case, we concluded that the endpoint ended up in a region where the model did not learn to decode well. This result indicates the transport learned by latent flow-matching is not absolutely invariant to changes, but remains stable as long as the latent coordinate system is fixed. VAE-swap did not degrade the performance like sign-flipping, it shows that changing the latent space alters the mapping to the endpoint latents. We will sharpen this point in the manuscript.
>
> (iii) **Fairness in the training budget**. Since we want to compare speeds of convergence across the different variants, we fixed the training budget (corresponding to the number of training iterations) across all variants so that it is identical. For example, on ImageNet, if a given training budget allows for only 200k iterations, then pr=0.95 would be beneficial. In contrast, if the budget allows for 400k iterations, then pr=0.75 is more beneficial.
>
> (iv) **Random pruning**. Yes, on CelebA-HQ, the pruning method matters more, its performance was worse than unpruned, gradient-based, loss-based and clustering-based pruning. On ImageNet, the pruning method is less critical (the variance is smaller than 1 FID unit) under a limited compute budget. We attribute this to characteristics of the data and how representative it is of the underlying distribution. If the dataset already provides a representative coverage of the underlying distribution, random sampling remains effective due to its i.i.d. nature.
>
> (vi) **Limitations**. We note that the phenomenon itself is a characteristic of LFM models behavior. Limitations related to the cost of the gradient-based approach for example is already included (line 188). Limitations pertaining to the proposed approach include also the lack of automatic adaptation for finding an optimal split point for the trajectory ($t_0$). We will include a dedicated paragraph on the limitations in the manuscript.

---

> > ### Author Rebuttal · Reviewer_CYev · 2026-04-03
> >
> > The rebuttal was helpful. The rebuttal helped clarify many of the concerns, especially regarding the fairness of the training comparison, the interpretation of pruning gains, and the latent perturbation results. The explanation for the dependence on the pruning rule on ImageNet is still somewhat high-level, but I do not see this as a major issue.
> >
> > Overall, the rebuttal resolved enough of my concerns, so I keep my score unchanged.

---

> > > ### Author Response · Authors · 2026-04-04
> > >
> > > Thank you for the helpful follow-up and for taking the rebuttal into account. We appreciate that the clarification resolved your main concerns.

---

### Official Review · Reviewer_KN24 · 2026-03-13

**Soundness:** 2
**Presentation:** 3
**Significance:** 2
**Originality:** 2
**Overall Recommendation:** 4
**Confidence:** 4

**Summary:**

This paper reports an stability phenomenon in Latent Flow Matching (LFM). Specifically, even when part of the training data is removed or the model size is changed, samples generated from the same initial noise tend to follow very similar generation trajectories and produce highly similar outputs. The authors argue that this phenomenon is tied to the properties of the Flow Matching objective itself, and that it can be explained through empirical observation.
The paper proposes two practical methods. First, it introduces data pruning, which aims to preserve performance without using the full dataset. When samples are selected for training based on criteria such as loss, gradients, and clustering, the balanced clustering strategy proves particularly effective. On ImageNet, the authors report that reducing the training data by as much as 75% still maintains performance while improving training efficiency. Second, the paper proposes a coarse-to-fine inference scheme, in which a smaller model handles the early part of the trajectory and a larger model refines the later stages. With this approach, they achieve about a 2.15× inference speedup without significantly degrading generation quality.
Experiments are conducted on CelebA-HQ, FFHQ, and ImageNet, and they consistently show that similar outputs are obtained from the same noise seed even under different data subsets or model conditions.

**Compliance With Llm Reviewing Policy:**

Affirmed.

**Final Justification:**

My final recommendation is weak accept. I found the paper to be original and empirically strong, with a convincing stability observation and a particularly interesting Coarse-to-Fine result that translates this insight into a practical acceleration method. My main concerns were not about soundness, but about how broadly the pruning results should be interpreted, especially since much of the evidence was in the compute-constrained or pre-convergence regime. The rebuttal addressed this point well by clarifying that the main claim is improved efficiency under realistic fixed-budget training rather than better asymptotic performance, and this clarification positively changed my evaluation.

**Key Questions For Authors:**

1. Do the benefits of dataset pruning persist at or near full convergence, or are they mainly a faster-convergence effect under a fixed training budget?
2. Given that the main ImageNet evaluations are performed in the pre-convergence regime, can the authors provide stronger evidence that performance and recall/coverage remain competitive after sufficiently long training?

**Limitations:**

yes

**Strengths And Weaknesses:**

## Strength
### Soundness
The paper’s central claims are supported by a broad and well-designed set of experiments. In particular, the authors do not stop at showing qualitative stability under matched noise seeds, but also test the phenomenon under dataset pruning, architectural changes, and model capacity variation, which makes the empirical case substantially more convincing. The experiments are also quite insightful in that they probe not only whether stability exists, but also when it breaks.

### Significance
I found the coarse-to-fine insight particularly interesting. The observation that early timesteps mainly capture coarse structure, making them amenable to a much smaller model, is a meaningful and nontrivial takeaway from the stability analysis.

## Weaknesses
### Significance
- While the pruning results are interesting, their practical significance is somewhat unclear. The method still assumes access to the full dataset and requires nontrivial preprocessing such as clustering or sample scoring. Therefore, the paper does not clearly quantify whether the additional subset-selection cost is offset by the downstream training savings in realistic large-scale settings.

### Evaluation
- The main pruning evaluation is largely conducted in the pre-convergence regime. In ImageNet, the key comparison across pruning fractions is made at 200k iterations, even though the longer training curves suggest that the gap can shrink substantially later in training. This makes the evidence stronger for faster convergence under a fixed budget than for improved final performance after sufficient training.
- The longer-horizon evaluation is also limited in scope. The paper notes that the longest ImageNet runs were only performed for \(C_b\), so it remains unclear whether the same conclusions hold more broadly across pruning strategies near convergence. The fact that extreme pruning can initially outperform and then degrade later already suggests that the conclusion should be stated more carefully.
- The diversity/coverage claim could be better supported. Although the appendix includes precision/recall-style metrics up to 600k iterations, the main text remains centered around FID and matched-seed similarity. These are useful indicators of trajectory consistency, but they are not equivalent to showing preservation of the full support of the learned distribution.
- The C2F result is compelling, but the evaluation would be stronger with broader comparisons against alternative acceleration baselines and a clearer analysis of sensitivity to architecture choices.

---

> ### Author Rebuttal · Authors · 2026-03-28
>
> We thank the reviewer for acknowledging our proposed Coarse-to-Fine approach is interesting and for raising these important questions, which we address below.
> (i) **Practical significance**. The preprocessing consists of a single forward pass over the dataset (for feature extraction) and clustering in latent space, and is therefore incurred only once. In contrast, training requires repeated forward and backward passes over many iterations. As a result, preprocessing is comparable to a small fraction of the overall training cost, while pruning reduces the effective number of iterations required to reach a given performance. In practice, even a modest reduction in training iterations is sufficient to offset the one-time preprocessing cost.
>
> In addition, preprocessing does not require annotations, such that pruning can reduce annotation effort in conditional setups, and it also reduces data loading overhead during training.
>
> (ii) **Benefit under a constrained budget**. Our paper makes two main observations: (a) LFM can be surprisingly robust against severe pruning in the convergence regime, especially when the right pruning methods are used (Fig 3). (b) in the computationally constrained regime, even random pruning can significantly improve image quality, yet if trained to convergence, the gap closes and, depending on the pruning ratio, can lead to significantly worse results such as pr=0.95 (cf. Fig 6).
> The reviewer is right that our strongest evidence supports improved training efficiency under a fixed compute budget rather than a uniformly better optimum after full convergence, and we will clarify this point in the manuscript. However, many frontier models are trained under compute-constrained budgets. Moreover, recent work on scaling laws for diffusion models shows that optimal performance must be understood under fixed compute budgets, where model size, data, and training cost must be balanced rather than maximized~\citep{liang2024scaling}. In this regime, improvements in optimization efficiency are often more relevant than asymptotic convergence. We would like to point out that the main finding of our work is that this stability enables improved efficiency under such compute-constrained regimes.
>
> (iii) **Generative metrics**. In the manuscript, we followed generative models practices in metric reporting which often focuses on FID due to space constraints. We acknowledge the limitations of a single metric, and therefore reported other distribution coverage metrics in the appendix. We note that on ImageNet, these distribution metrics are preserved despite pruning (cf Fig. 10). We will include these metrics in the revised version of the manuscript.
>
> (iv) **Other acceleration approaches**. Existing approaches such as Blockwise Flow Matching [\*] or distillation (teacher-student) methods such as SlimFlow [**] are related but orthogonal to our approach. For instance, unlike [\*,\*\*], our approach does not require training specialized architectures or modifying the FM objective. Furthermore, our approach is general and could be combined with existing approaches. Based on our analytical analysis and experiments with the U-Net architecture (Fig 4.a), the approach is not tied to a specific architecture, only stability is critical for its performance as shown in the paper (line 342-348). The approach still yields competitive performance even without finetuning; finetuning improves it only slightly. We will include this comparison in the paper.
>
> \* Blockwise Flow Matching: Improving Flow Matching Models For Efficient High-Quality Generation, Park et al.
>
> \*\* Slimflow: Training smaller one-step diffusion models with rectified flow, Zhu et al.

---

> > ### Author Rebuttal · Reviewer_KN24 · 2026-04-04
> >
> > I thank the authors for the detailed rebuttal and clarifications. In particular, I appreciate the clarification that the main contribution is not necessarily that pruning yields a uniformly better optimum after full convergence, but rather that the observed stability can be exploited to improve training efficiency under compute-constrained regimes. Since my main concern was precisely about how to interpret the pruning results under limited training budgets, this response addresses an important part of my evaluation. I will therefore raise my evaluation accordingly.

---

> > > ### Author Response · Authors · 2026-04-04
> > >
> > > Thank you for the thoughtful follow-up and for reconsidering your evaluation. We appreciate that the clarification of the compute-constrained setting addressed your main concern.

---

### Decision · Program_Chairs · 2026-04-30

**Decision:**

Accept (regular)

**Comment:**

This submission make the empirical observation that when flow matching is performed in the latent space of an encoder-decoder (such as a VQVAE), the learned flow is robust to operations like pruning (removing data points from the training set) and model capacity reduction. The authors attribute this phenomenon to the fact that the target velocity in a flow matching problem can be expressed in closed form as a weighted sum over training data points, where for many higher dimensional dataset, the entropy over weights collapses relatively early on in interpolation time, after which one sample dominates in the sum. They show that this stability can be used to achieve faster training (by pruning the dataset) and that it is also possible to employ what they refer to as a coarse-to-fine (C2F) inference scheme that combine a small model for early steps with a larger model for later ones, which achieves a speedup at inference time. Experiments are conducted on CelebA-HQ, FFHQ, and ImageNet.

Reviewers are overall positive about this submission. Reviewers note that the submission is clearly written, with good figures. The also find the observation about stability interesting and nontrivial, find the experiments well-designed, and find the coarse-to-fine contribution an interesting way to speed up inference. Noted weaknesses focus on the fact that some claims are stronger than the evidence. The pruning results are evaluated pre-convergence, which means that they are more a measure of convergence under a fixed training budget than of stability under removal of data. One reviewer also notes that the speedup in C2F now requires two models and manual tuning of the cross-over time between the two. During discussion, authors added CelebA-HQ (512x512) results, a coreset baseline, evaluation on the HumanML3D motion dataset and clarified various aspects of the exposition. One reviewer raised their score and all reviewers marked concerns partially or fully resolved.

The area chair would like to add some observations and suggestions beyond those raised by reviewers. The paper's framing that one training sample dominates seems slightly imprecise. At time 0 the weights in equation (2) are uniform, which means that all trajectories initially converge on the mean of the data distribution. Then there is a cross-over regime, after which trajectories indeed become dominated by a single sample. This interpretation seems entirely compatible with the C2F approach proposed by the authors. More generally it is not entirely clear to the area chair where the latent space factors into the narrative. There is a bit of discontinuous jump from the observations about closed-form flows in section 2 to the implications of these observations to flows in latent spaces. More broadly it is not entirely clear whether it matters if the latent space is quantized or not. In this context, some experiments with a frozen continuous KL-VAE, as well as experiments in pixel space, could be informative. The area chair would hypothesize that generation of high quality samples requires a frozen decoder, but is less sure about the observed stability of trajectories.

Overall this seems a submission that, while perhaps not fully coherently formulated, makes interesting observations backed up by reasonable experiments, and which is above the bar for acceptance.